# Sox5 regulates beta-cell phenotype and is reduced in type 2 diabetes

A.S. Axelsson[1], T. Mahdi[1,2], H.A. Nenonen[1], T. Singh[1], S. Hänzelmann[3], A. Wendt[1], A. Bagge[1], T.M. Reinbothe[1], J. Millstein[4], X. Yang[4,5], B. Zhang[4,6], E.G. Gusmao[3], L. Shu[5], M. Szabat[7], Y. Tang[1,8], J. Wang[1,9], S. Salö[1], L. Eliasson[1], I. Artner[1], M. Fex[1], J.D. Johnson[7], C.B. Wollheim[1,10], J.M.J. Derry[4], B. Mecham[11], P. Spégel[1,12], H. Mulder[1], I.G. Costa[3], E. Zhang[1] & A.H. Rosengren[1,4,13]

Type 2 diabetes (T2D) is characterized by insulin resistance and impaired insulin secretion, but the mechanisms underlying insulin secretion failure are not completely understood. Here, we show that a set of co-expressed genes, which is enriched for genes with islet-selective open chromatin, is associated with T2D. These genes are perturbed in T2D and have a similar expression pattern to that of dedifferentiated islets. We identify *Sox5* as a regulator of the module. *Sox5* knockdown induces gene expression changes similar to those observed in T2D and diabetic animals and has profound effects on insulin secretion, including reduced depolarization-evoked $Ca^{2+}$-influx and β-cell exocytosis. *SOX5* overexpression reverses the expression perturbations observed in a mouse model of T2D, increases the expression of key β-cell genes and improves glucose-stimulated insulin secretion in human islets from donors with T2D. We suggest that human islets in T2D display changes reminiscent of dedifferentiation and highlight *SOX5* as a regulator of β-cell phenotype and function.

[1] Lund University Diabetes Center, CRC 91-11 SUS, Jan Waldenströms gata 35, SE-20502 Malmö, Sweden. [2] Medical Research Center, Hawler Medical University, 44001 Erbil, Iraq. [3] Institute of Biomedical Engineering, RWTH Aachen University Hospital, Pauwelstr 19, 52074 Aachen, Germany. [4] Sage Bionetworks, 1100 Fairview Avenue N, Seattle, Washington 98109, USA. [5] Department of Integrative Biology and Physiology, University of California, Los Angeles, 610 Charles E. Young Dr East, Los Angeles, California 90095, USA. [6] Department of Genetics and Genomic Sciences, Icahn Institute of Genomics and Multiscale Biology, Icahn School of Medicine at Mount Sinai, 1470 Madison Avenue, New York, New York 10029, USA. [7] Diabetes Research Group, Department of Cellular and Physiological Sciences, Life Sciences Institute, University of British Columbia, 5358–2350 Health Sciences Mall, Vancouver, British Columbia, Canada V6T 1Z3. [8] Key Lab of Hormones and Development, Ministry of Health, Metabolic Diseases Hospital, Tianjin Medical University, Tianjin 300070, China. [9] Department of Emergency, Zhongshan Hospital, Xiamen University, Xiamen, Fujian 361004, China. [10] Department of Cell Physiology and Metabolism, University Medical Center, Rue Michel-Servet 1, 1206 Geneva, Switzerland. [11] Trialomics, 6310 12th Avenue NE, Seattle, Washington 98115, USA. [12] Centre for Analysis and Synthesis, Department of Chemistry, Lund University, SE-221 00 Lund, Sweden. [13] Department of Neuroscience and Physiology, University of Gothenburg, Box 100, SE-405 30 Gothenburg, Sweden. Correspondence and requests for materials should be addressed to A.H.R. (email: anders.rosengren@gu.se).

Type 2 diabetes mellitus (T2D) results from a combination of insufficient insulin secretion from the pancreatic islets and insulin resistance of target cells[1].

Pancreatic β-cell mass is reduced by ∼50% in individuals with T2D compared with non-diabetic subjects[2,3]. However, glucose-stimulated insulin secretion is decreased in isolated islets from human donors with T2D, even after correction for insulin content, suggesting an important role also of functional defects[4–6].

In the β-cell, glucose metabolism leads to increased cytosolic ATP, closure of ATP-sensitive $K^+$ channels ($K_{ATP}$-channels), initiation of electrical activity and $Ca^{2+}$-dependent exocytosis of insulin-containing granules[7]. Despite the extensive characterization of the secretory process in normal β-cells, the mechanisms that lead to β-cell failure in T2D remain largely unknown.

Recent genome-wide association studies have identified more than 80 loci associated with T2D risk[6]. Furthermore, global gene expression studies have identified a plethora of genes that are differentially expressed in islets from T2D donors compared with control subjects[7,8]. However, these large-scale data have not yet been maximally utilized to identify pathophysiological mechanisms.

Network models have been proposed as a useful framework for studying complex data[9]. To take full advantage of such models to provide pathophysiological insights and identify new disease genes for T2D, it is important to combine bioinformatics with detailed cellular investigations, as has recently been demonstrated[10,11].

To investigate the defects that lead to β-cell failure in T2D, we analysed the co-expression networks of human pancreatic islets. We identified a set of co-expressed genes ('module') that is associated with T2D and reduced insulin secretion and show that human islets display expression perturbations reminiscent of β-cell dedifferentiation. The data also highlight Sox5 as a previously unrecognized regulator of β-cell gene expression and secretory function.

## Results

**A gene co-expression module associated with T2D.** We first obtained global microarray expression data from islets from 64 human donors, of which 19 had T2D (Supplementary Table 1), and explored gene co-expression using the weighted gene co-expression network analysis (WGCNA) framework[12] (see Experimental Procedures). First, we calculated the connectivity, reflecting the extent of co-expression for all pairs of gene expression traits (Supplementary Table 2). We then used the topological overlap, which for each gene pair measures the number of similar connections of the two genes with all other genes in the array, to identify 56 gene co-expression modules (Fig. 1a).

Rather than analysing each gene individually, we used the first principal component of the gene expression traits of each module (the 'module eigengene', which reflects a summary expression of all module genes). One eigengene, representing a module with 3,032 genes (module 2 in Supplementary Table 3, nominal $P$ values), stood out as being correlated with both T2D status ($P = 0.01$; logistic regression; $n = 64$) and HbA1c ($P = 0.003$; linear regression; $n = 52$), as well as insulin secretion in response to 16.7 mM glucose ($P = 0.006$; linear regression; $n = 48$) and 70 mM $K^+$ ($P = 0.048$; linear regression; $n = 26$). Henceforth, we refer to this as the T2D-associated module.

The T2D-associated module was enriched for genes known to be highly expressed in the pancreas (1.8-fold enrichment; $P < 1E-20$ using Fisher's exact test) and genes involved in vesicle release (1.6-fold; $P = 3E-9$; Fisher test) and secretory function

(2.1-fold; $P = 4E-6$; Fisher test). On the basis of these data, we hypothesized that the gene module contributes to the maintenance of β-cell function and tested the hypothesis by interrogating data on regulatory DNA in human islets[13,14]. Gaulton and colleagues recently identified 340 genes located to regions with islet-selective open chromatin[13]. The T2D-associated module that we identified contained 168 of those genes (Supplementary Table 4), corresponding to 49% of all genes with islet-selective open chromatin (only 14% expected by chance; $P < 1E-6$; Fisher test).

The eigengene representing the 168 genes had an even stronger association with T2D-related traits than that of the entire module and was correlated with diabetes status, HbA1c, and glucose-stimulated and $K^+$-stimulated insulin secretion (Fig. 1b–e). When analysing only non-diabetic donors ($n = 45$), there was no association between the eigengene and insulin secretion among those with BMI below median ($27\,kg\,m^{-2}$). By contrast, among non-diabetic donors with BMI above median, which may be at higher risk of developing T2D, the eigengene was associated with impaired glucose-stimulated insulin secretion ($P = 0.04$; $\beta = 1.36$; linear regression).

Co-expression networks typically have a few highly linked hubs that connect a large number of peripheral nodes[9]. We computed the total connectivity of each gene (the degree $k_{in}$, which reflects the number of genes to which it is linked in the module) and observed a significant correlation between $k_{in}$ and the gene expression association with T2D ($P < 1E-6$, Pearson correlation $r = 0.35$). The $k_{in}$ of the 168 open chromatin genes was on average 50% higher than the overall connectivity of the module genes ($P < 1E-6$; Fisher test; Fig. 1f). These data suggest that the 168 open chromatin genes are at the core of the T2D-associated module and may play an important role in maintaining normal secretory function.

We replicated the analyses in an additional 59 human donors (22 with T2D; Supplementary Tables 5 and 6), and identified 24 co-expression modules. The eigengene displaying the strongest association with T2D and HbA1c in the replication set (nominal $P = 0.06$ for T2D and $P = 0.06$ for HbA1c, one-sided regression) represented a module of 2,439 genes. This module had a large overlap with the T2D-associated module identified in the initial analysis (1,198 overlapping genes; $P < 1E-20$; Fisher test). Of the 168 genes with islet-selective open chromatin in the initial module, 90% (152 genes) were also present in the replication module ($P < 1E-20$; Fisher test; Supplementary Table 7). The eigengene of these genes had lower values in T2D islets ($P = 0.04$, one-sided logistic regression), was negatively correlated with HbA1c ($P = 0.04$, one-sided linear regression), and exhibited a twofold higher connectivity ($k_{in}$) compared with the other module genes ($P < 1E-20$; Fisher test), which further corroborates the initial analyses.

**The T2D gene signature is reminiscent of immature β-cells.** We next explored the possibility of using the expression profile of the 168 open chromatin genes in diabetic versus non-diabetic donors as a 'T2D signature' of human islets to learn more about the associated pathophysiology. The signature was compared with gene expression profiles from >8,100 publically available microarray data sets. The expression profile exhibiting the highest overlap with the T2D signature was from artificially dedifferentiated human islets (GSE15543) (ref. 15) (130 genes in common; $P = 3E-68$; Fisher test; Fig. 1g). The data sets exhibiting the second and third highest overlap compared mature islets versus a fetal pancreatic cell line (82 overlapping genes; $P = 1E-19$; GSE18821) and mature islets versus early islet progenitors (81 overlapping genes; $P = 0.0003$; GSE23752),

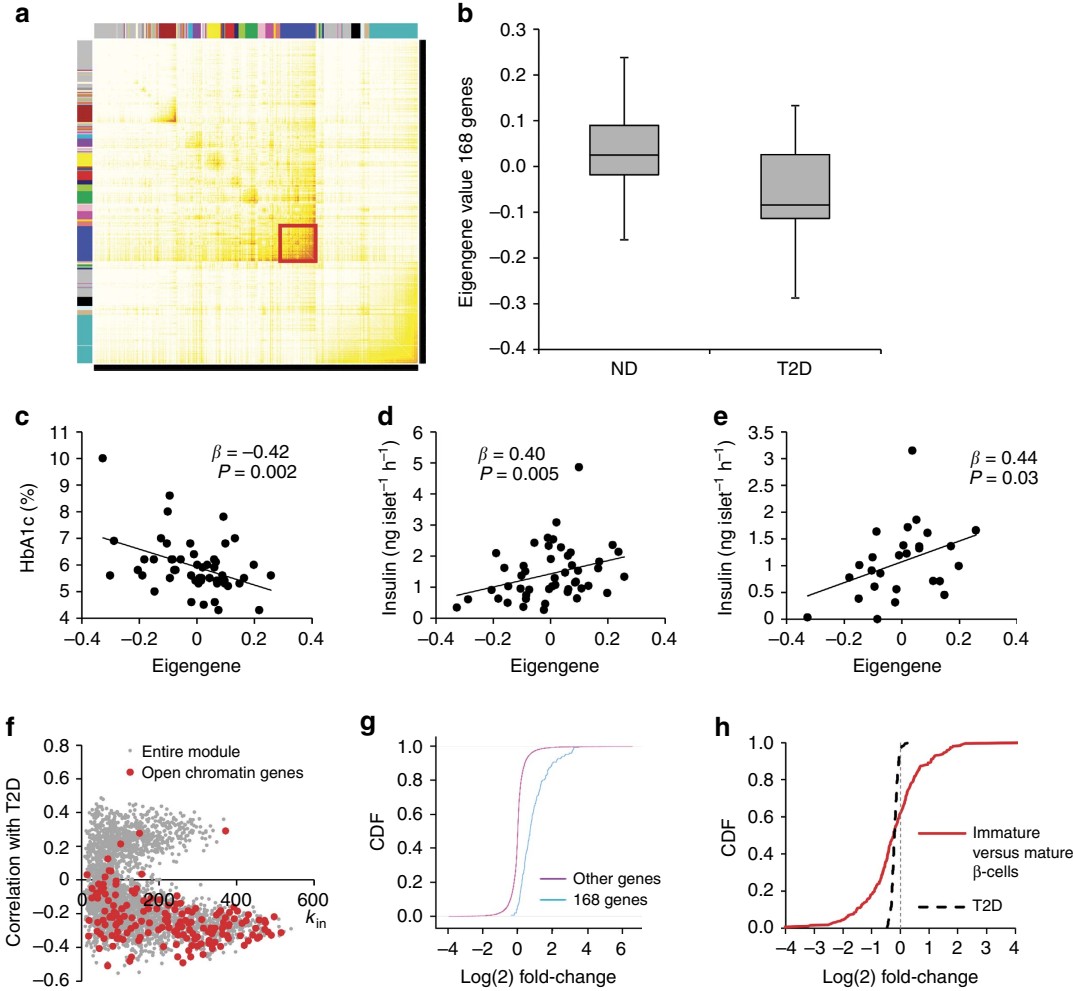

**Figure 1 | Co-expression network analysis and association between eigengene and type 2 diabetes traits.** (**a**) Symmetrically arranged heatmap of the topological overlap matrix for which the rows and columns are sorted by the hierarchical clustering tree used to define modules. The red square denotes the T2D-associated co-expression module. (**b**) Box plot showing the value of the eigengene for the 168 open chromatin genes in islets from non-diabetic (ND; $n = 45$) and T2D donors ($n = 19$). The box shows averages ± s.e.m. and the error bars denote the tenth and ninetieth percentile, respectively. (**c–e**) The module eigengene of the 168 open chromatin genes displayed against HbA1c (**c**; $n = 52$), glucose-stimulated insulin secretion (**d**; $n = 48$) and K$^+$-stimulated insulin secretion (**e**; $n = 26$). Statistical comparisons using linear regression. (**f**) The connectivity $k_{in}$ of each gene is displayed against the $r$ value for the Pearson correlation between the gene expression trait and T2D status. Grey dots denote genes in the T2D-associated module and red dots denote genes with islet-selective open chromatin. Data are from human islets from 64 donors. (**g**) Cumulative density function (CDF) plots of log2-transformed gene expression fold-change in freshly isolated versus expanded islets in microarrays from GSE15543. The blue line denotes the fold-change of the 168 open chromatin genes in GSE15543 and the purple line denotes the fold-change of the remaining genes in the array. (**h**) CDF plot of log2-transformed expression fold-change of genes in the T2D signature in Pdx1$^+$/Ins$^{low}$ (immature) versus Pdx1$^{high}$/Ins$^{high}$ (mature) human β-cells. The CDF plot of the 168 signature genes in T2D islets is also displayed.

respectively. Isolated human islet cultures may contain a large number of non-endocrine cells, and the nature of the expanded islet cells in GSE15543 is not fully characterized. Therefore, we compared the T2D signature with expression data on purified human β-cell fractions with low insulin expression (Pdx1$^+$/Ins$^{low}$), suggestive of immature β-cells, and high insulin expression (mature β-cells; Pdx1$^{high}$/Ins$^{high}$), respectively[14,16]. The T2D signature genes had higher expression in the purified mature β-cell fraction versus unpurified intact islets (1.7-fold enrichment; $P < 1E-6$; Fisher test) and was perturbed in immature versus mature human β-cells in a similar manner to that observed in T2D versus non-diabetic islets (1.3-fold enrichment; $P < 1E-6$; Fig. 1h). Taken together, these findings suggest that perturbation of the T2D module is associated with an immature β-cell state and loss of secretory function. In support of this notion, β-cell dedifferentiation has been shown to account for diabetes in

animals that are under physiological stress such as ageing and multiparity[17].

**Putative key regulators of the T2D-associated module.** We next aimed to identify key regulators of the T2D module that could have a pathogenetic role in β-cell failure. First, we analysed transcription factor binding sites (TFBS) for the 168 genes and found a high enrichment of putative binding sites for *SOX5* ($P = 1E-10$; Fisher test) and *TCF3* ($P = 4E-10$). Second, we found a significant overlap between the T2D signature and the expression changes induced by knockdown of *RORB* (GSE16585; $P = 0.0018$; Fisher test), *GRN* (GSE13162; $P = 1E-16$) and *PTCH1* (GSE24628; $P = 2E-19$), respectively, and over-expression of *LPAR1* (GSE15263; $P = 2E-11$). Third, we found that single nucleotide polymorphisms (SNPs) near *SMARCA1*

and *SOX5* were associated with the module eigengene (Supplementary Table 8). Fourth, we identified *TMEM196* ($P < 1E-6$; Pearson correlation $r = 0.85$) and *TMEM63C* ($P < 1E-6$; $r = 0.82$) as being the most highly correlated genes with the module eigengene of the 168 open chromatin genes.

We next analysed the expression of these putative regulators after stressing rat islets with 20 mM glucose or 1 mM palmitate (Supplementary Fig. 1a). We observed a protracted reduction of *Sox5*, *Lpar1*, *Ptch1*, *Smarca1*, *Tcf3*, *Tmem196* and *Tmem63C* mRNA. We also observed a > 50% decrease of mRNA levels of *Pdx1* and *Mafa* and a ~10-fold elevation of *Ldha*, which is normally low in β-cells. These expression changes are reminiscent of what has previously been observed in immature β-cells[18,19], although it should be stressed that the 48 h incubation is far different from the prolonged dysmetabolic state characterizing T2D.

To directly assess the function of these genes, each of the putative regulators was silenced using siRNA in clonal rat INS-1 832/13 cells (Fig. 2a). *Sox5* silencing ($72 \pm 2\%$) reduced glucose-stimulated insulin secretion by 50% ($P = 0.003$), whereas silencing of the other genes had no significant effect on insulin secretion (using two different siRNAs for each gene). Sox5 silencing was repeated using a total of five different siRNAs in separate experiments, with two of them causing a significant reduction of glucose-stimulated insulin secretion, one causing a 20% increase and two being without a significant effect (Supplementary Fig. 1b). The siRNAs that decreased insulin secretion target different parts of the *Sox5* mRNA sequence corresponding to the conserved and functionally important first coil region of the protein, while the siRNAs with no or stimulatory effect on insulin secretion target sequences outside of the coil region.

### Sox5 knockdown impairs glucose-stimulated insulin secretion.

*SOX5* encodes sex determining region Y-box 5, a transcription factor involved in chondrogenesis and neurogenesis[20]. *SOX5* lacks a transactivation domain but binds close to other transcription factors, suggesting that it orchestrates the chromatin structure[20]. *SOX5* mRNA is expressed both in purified human α- and β-cell fractions and to a smaller extent in the exocrine pancreas[7]. To date *SOX5* has not been implicated in β-cell function or T2D.

We first analysed *SOX5* mRNA levels in human islets and observed reduced *SOX5* expression in islets from T2D donors compared to non-diabetic controls ($P = 0.04$; Supplementary Fig. 1c). Immunohistochemistry of pancreatic sections showed that SOX5 protein was expressed both in α- and β-cells, and, to a lower degree, in exocrine tissue. SOX5 was present both in the nucleus and in the cytosol, and cytosolic expression was especially evident in α-cells. Nuclear SOX5 was reduced by 67% in T2D β-cells compared with non-diabetic β-cells ($P < 0.001$; Supplementary Fig. 2).

We next knocked down *Sox5* (*Sox5*-kd) in INS-1 832/13 cells ($76 \pm 4\%$ knockdown; $P = 1E-6$; Student's *t*-test was used in all cellular experiments unless otherwise specified), which caused decreased mRNA expression of *Pdx1* (14%; $P = 0.0005$) and *Mafa* (52%; $P = 0.0004$) relative to cells treated with a negative control siRNA (Fig. 2b). Immunostainings showed a reduction of MAFA expression in *Sox5*-kd cells, while MAFB, NKX6.1 and PAX6 were unaffected (Supplementary Fig. 3a). MAFA ($P = 0.01$; Supplementary Fig. 3b) but not PDX1 protein levels were reduced in *Sox5*-kd cells as assessed by Western blot.

At low glucose levels, insulin secretion in *Sox5*-kd cells was similar to control cells (Fig. 2c). By contrast, at 5 mM glucose and above we observed a significant ~50% reduction of insulin

secretion after *Sox5*-kd. Notably, the ratio between secreted proinsulin versus insulin was 73% higher in *Sox5*-kd cells at 2.8 mM glucose ($P = 0.005$) and 96% higher at 16.7 mM glucose ($P = 0.01$; Fig. 2d). The processing of proinsulin to insulin has been shown to be impaired in T2D (ref. 21). The overall viability was similar between *Sox5*-kd and control cells (Supplementary Fig. 3c). There was no difference in insulin content between *Sox5*-kd and control cells (Supplementary Fig. 3d). Accordingly, glucose-stimulated insulin secretion was significantly reduced in *Sox5*-kd cells also after normalization for insulin content (Supplementary Fig. 3e).

To further define the mechanistic defects caused by *Sox5* downregulation we treated cells with each of the mitochondrial substrates leucine and α-ketoisocaproic acid (α-KIC), the $K_{ATP}$ channel inhibitor tolbutamide, and high $K^+$ to directly depolarize the β-cells. The fold-stimulation of insulin secretion was reduced in response to all these secretagogues in *Sox5*-kd cells (Fig. 2e,f). Glucose-mediated amplification of insulin secretion distal to $K_{ATP}$ channel closure (Supplementary Fig. 4a) and the responses to glucagon-like peptide-1 and the α2-adrenergic receptor agonist clonidine were unaffected (Supplementary Fig. 4b). Intracellular concentrations of cAMP were similar in *Sox5*-kd and control cells (Supplementary Fig. 4c).

We also observed that *SOX5* expression in human islets was associated with both HbA1c ($\beta = -0.33$; $P = 0.04$; linear regression; $n = 123$) and insulin secretion in response to high glucose ($\beta = 0.22$; $P = 0.02$; $n = 48$) and high $K^+$ ($\beta = 0.93$; $P = 0.02$; linear regression; $n = 26$). There was no effect of sex on any of these associations. When analysing human β-cells from six diabetic donors by electron microscopy, we found an association between decreased islet expression of *SOX5* and reduced numbers of insulin granules docked at the plasma membrane ($P = 0.01$; linear regression; Fig. 2g and Supplementary Fig. 4d). There was no association between *SOX5* expression and docked granules in non-diabetic donors ($P = 0.5$; $n = 11$ donors). By contrast, in human α-cells decreased islet expression of *SOX5* was rather associated with increased number of docked granules (Supplementary Fig. 4e). It should be noted that the changes in the number of docked granules in T2D donors with low or high *SOX5* expression in islet is correlative, as these cells may have reduced levels of several islet-enriched transcription factors.

### Sox5 knockdown impairs glucose metabolism.

There was a 50% decrease in mitochondrial oxygen consumption rate (OCR) in response to glucose ($P = 0.02$) or pyruvate ($P = 0.001$) in *Sox5*-kd cells (Fig. 3a), paralleled by an accumulation of early glycolytic intermediates and a reduction of the Krebs cycle intermediate fumarate ($P = 0.046$; Fig. 3b). Alanine ($P = 0.04$) and lactate ($P = 0.009$), which are both generated from pyruvate, were elevated in *Sox5*-kd cells relative to control cells at high glucose (Fig. 3c). These findings point to a clear metabolic defect in *Sox5*-kd cells. This defect is unlikely to involve the glycolytic machinery because OCR in response to pyruvate (that bypasses glycolysis) was reduced to a similar extent as in response to glucose. Rather, the data suggest a perturbation in mitochondrial shuttles and a shift in the balance between aerobic and anaerobic metabolism.

### Sox5 knockdown reduces expression of L-type $Ca^{2+}$ channels.

We further explored the secretion defect in *Sox5*-kd cells by measurements of cell capacitance to monitor the exocytotic capacity. Total exocytosis in response to a train of ten depolarizations tended to be reduced (by 28%) in *Sox5*-kd cells, but the decrease was not statistically significant ($P = 0.07$; Fig. 4a). However, analysis of rapid exocytosis (estimated as the response

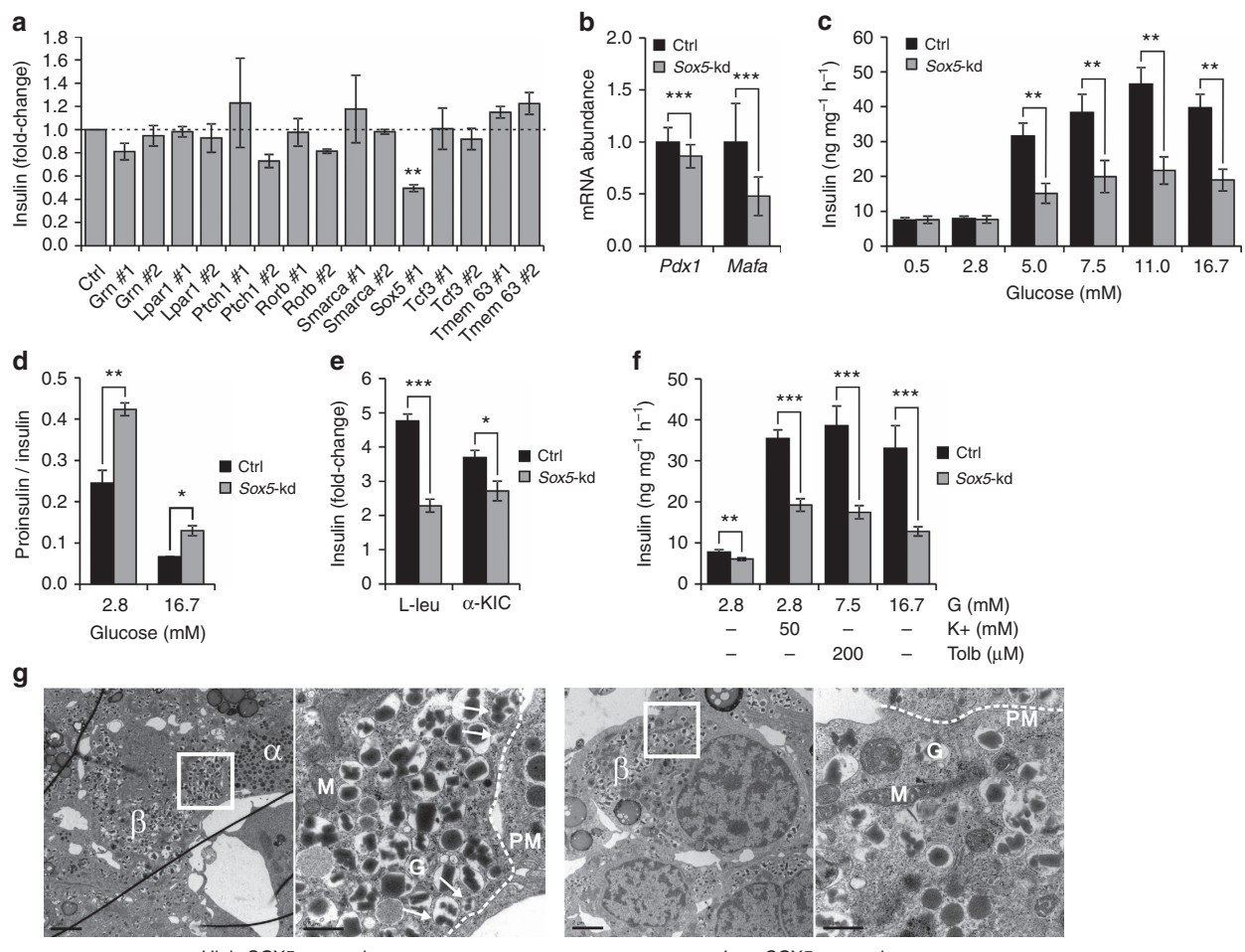

**Figure 2 | Characterization of *Sox5* expression and effects of *Sox5* knockdown.** The experiments were performed in INS-1 832/13 cells if not stated otherwise. *Sox5*-kd cells were transfected with Sox5 siRNA 48 h before the experiment, and control cells were transfected with negative control siRNA. (**a**) Fold-change of insulin secretion at 16.7 mM glucose compared to control-treated cells in response to gene silencing. Data for each siRNA is compared with cells treated with a negative control siRNA (*n* = 3–5 experiments per siRNA). *P* values are corrected for multiple comparisons. *Tmem196* was excluded due to undetectable expression. (**b**) Expression of *Pdx1* and *Mafa* mRNA following *Sox5* knockdown (*Sox5*-kd) relative to control cells (*n* = 8). (**c**) Secreted insulin normalized to cellular protein content during a 1-h stimulation with glucose as indicated (*n* = 4). (**d**) Ratio of secreted proinsulin to insulin following a 1-h stimulation with 2.8 or 16.7 mM glucose (G) in *Sox5*-kd and control cells (*n* = 4 samples in 1 experiment). (**e**) Fold-change of insulin secretion in *Sox5*-kd and control cells after a 1-h incubation at 2.8 mM glucose with 10 mM L-leucine (L-leu) or 10 mM α-ketoisocaproic acid (α-KIC) relative to 2.8 mM glucose (*n* = 3 per condition). (**f**) Insulin secretion normalized to cellular protein content in *Sox5*-kd and control cells in response to glucose (G), 50 mM K$^+$ and 200 μM tolbutamide (tolb) as indicated (*n* = 3 per condition). (**g**) Representative electron micrographs of human islets from T2D donors with high or low *SOX5* expression (5–9 cells analysed from each of 6 donors with T2D). Mitochondria (M), plasma membrane (PM), insulin granules (G), nucleus (N) and α- and β-cells are indicated. Granules were defined as docked (indicated by arrows) when located within 150 nm from the PM. Scale bar, 2 μm in the lower and 0.5 μm in the larger magnification (corresponding to the marked area). Data are mean ± s.e.m. *$P<0.05$; **$P<0.01$; ***$P<0.001$ using Student's *t*-test.

to the first two depolarizations) and slow exocytosis (the response to pulses 3–10), proposed to correlate with 1st- and 2nd-phase insulin secretion[22], revealed a distinct (44%) reduction of rapid exocytosis in *Sox5*-kd cells ($P = 0.02$; Fig. 4b). Interestingly, the first phase of insulin secretion has been suggested to be perturbed in T2D patients[23]. There was a parallel reduction of the integrated Ca$^{2+}$ current in *Sox5*-kd cells ($P = 0.002$) (Fig. 4c). By contrast, the Ca$^{2+}$ sensitivity (Fig. 4d) and the exocytotic rate (Fig. 4e) were similar between *Sox5*-kd and control cells, indicating that the exocytotic machinery was intact. There was no difference in K$_{ATP}$-channel conductance between *Sox5*-kd and control cells, suggesting that K$_{ATP}$-channel activity was not perturbed (Supplementary Fig. 4f).

To specifically study the effect of *Sox5* knockdown on glucose-induced changes in intracellular Ca$^{2+}$ ([Ca$^{2+}$]$_i$), we conducted measurements with the Ca$^{2+}$ sensor Fluo-5F (Fig. 4f). The increase in [Ca$^{2+}$]$_i$ elicited by an elevation of glucose from 2.8 to 20 mM was reduced in *Sox5*-kd cells ($P = 0.04$ for area under the curve). Moreover, analysis of the current–voltage relationship revealed a pronounced reduction of the peak Ca$^{2+}$ current in *Sox5*-kd (Fig. 4g). Blocking of L-type Ca$^{2+}$ channels by isradipine evoked a significant reduction of the peak Ca$^{2+}$ current in control cells. By contrast, in *Sox5*-kd cells isradipine had no further inhibitory effect on the peak Ca$^{2+}$ current, showing that the L-type component of the Ca$^{2+}$ current was affected by *Sox5*-kd (Fig. 4g). This was corroborated by immunostaining, which demonstrated a ∼20% reduction of the expression of L-type Ca$^{2+}$ channels ($P = 0.04$ for Ca$_V$1.2 and $P = 0.06$ for Ca$_V$1.3) in *Sox5*-kd cells relative to control cells (Supplementary Fig. 5a), and by Western blot, which showed that

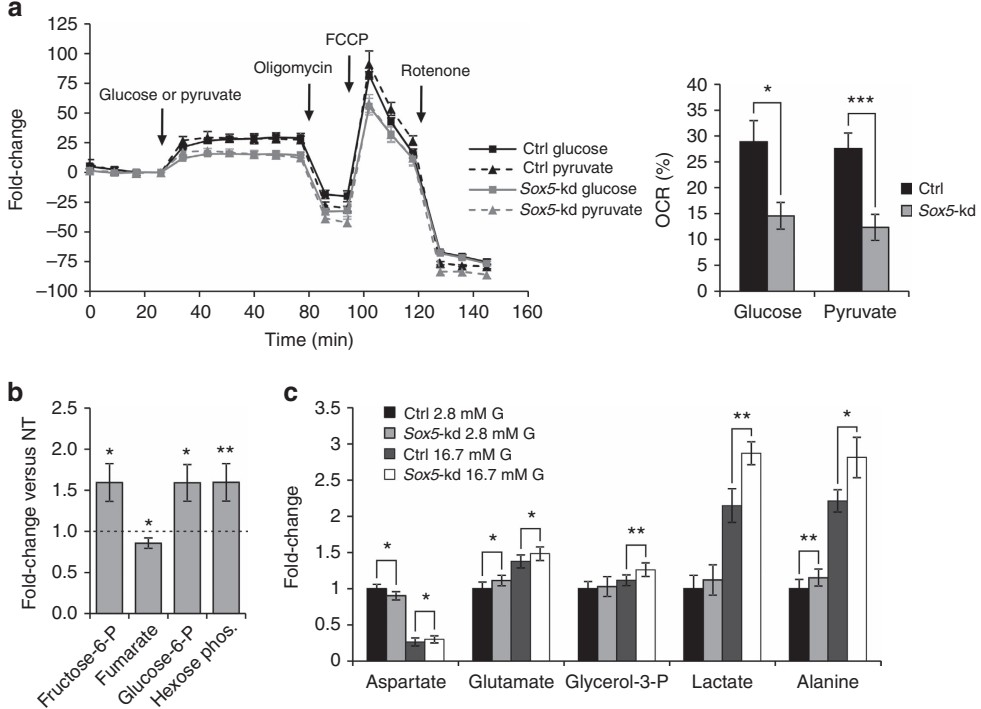

**Figure 3 | Metabolic characterization of *Sox5*-kd cells.** The experiments were performed in INS-1 832/13. *Sox5*-kd cells were transfected with Sox5 siRNA 48 h before the experiment, and control cells were transfected with negative control siRNA. (**a**) Mitochondrial oxygen consumption rate (OCR) measured by Seahorse XF24. Data in each group are normalized to the values at 2.8 mM glucose (*n* = 3). Glucose, pyruvate, oligomycin (which inhibits ATP synthase in order to assess the mitochondrial proton leak), the uncoupler FCCP (which maximizes respiration) and rotenone (which inhibits the electron transport chain) were added as indicated. The bar graph shows average mitochondrial OCR in control (Ctrl) and *Sox5*-kd cells (*n* = 3). (**b**) Fold-change of glucose-6-phosphate (Glucose-6-P), fructose-6-phosphate (Fructose-6-P), hexose phosphates and fumarate at 2.8 mM glucose in *Sox5*-kd cells versus non-treated control cells (NT) measured by gas chromatography/mass spectrometry (*n* = 5). (**c**) Levels of aspartate, glutamate, glycerol-3-phosphate (Glycerol-3-P), lactate and alanine at 2.8 and 16.7 mM glucose in *Sox5*-kd and control cells. Data are normalized to control cells at 2.8 mM glucose. (*n* = 5) Data are mean ± s.e.m. *P<0.05; **P<0.01; ***P<0.001 using Student's *t*-test.

$Ca_V1.2$ was reduced by 20% ($P = 0.03$) and $Ca_V1.3$ by 15% ($P = 0.02$; Supplementary Fig. 5b). The L-type $Ca^{2+}$ channels $Ca_V1.2$ (*CACNA1C*) and $Ca_V1.3$ (*CACNA1D*) are contained in the T2D-associated module and the expression of these genes was strongly correlated with *SOX5* in human islets (Pearson correlation $r = 0.85$; $P < 1E-6$ for *CACNA1C* and $r = 0.82$, $P < 1E-6$, for *CACNA1D*; $n = 123$). These findings demonstrate that *Sox5*-kd cells, in addition to pronounced metabolic defects, have reduced L-type $Ca^{2+}$ channels, decreased depolarization-evoked $Ca^{2+}$ influx and consequently reduced insulin exocytosis.

These observations were paralleled by patch clamp recordings of human β-cells from 18 non-diabetic donors, which demonstrated an association between *SOX5* islet expression and β-cell exocytosis ($P = 0.03$ using one-sided linear regression). There was also an association between *SOX5* expression and the integrated $Ca^{2+}$ current in response to the first depolarization in human β-cells ($P = 0.04$ using one-sided linear regression). Next, we transfected dispersed human β-cells with siRNA targeting *SOX5*. The cells were co-transfected with Alexa555-coupled oligonucleotides to enable recordings of cells that were fluorescent, indicating effective transfection. There was a 49% reduction of total exocytosis following this treatment ($P = 0.04$, one-sided *t*-test; Fig. 4h). Moreover, early exocytosis was reduced by 54% compared with control cells ($P = 0.04$), which corroborates the findings in INS-1 832/13 cells.

**Overexpression of *Sox5* improves insulin secretion.** We transiently overexpressed *Sox5* by co-transfecting INS-1 832/13

cells with Sox5- and GFP-expressing plasmids. Capacitance measurements of GFP-fluorescent cells showed a 72% increase in exocytosis relative to control cells (Supplementary Fig. 5c). Rapid exocytosis was particularly pronounced (Supplementary Fig. 5d) and the integrated $Ca^{2+}$ current was increased (Supplementary Fig. 5e), which mirror the results observed in *Sox5*-kd cells. Protein levels of $Ca_V1.2$ were increased by 30% ($P = 0.0076$; Supplementary Fig. 5f) as assessed by Western blot. We next overexpressed *Sox5* in intact rat islets using lentivirus infection. Dispersion of transduced islets to single cells and separation of the β-cell fraction using fluorescence-activated cell sorting followed by PCR showed higher absolute values of *MafA* mRNA in cells with *Sox5* overexpression (100- to 600-fold *Sox5* increase) compared with control cells, without reaching statistical significance (Supplementary Fig. 5g). We also analysed the pure β-cell fraction compared with the β-cell-depleted fraction (*Ins1* expression in the latter fraction was ∼30% of that in the pure fraction). We observed an increase in *Pdx1* expression in the pure β-cell fraction compared to the β-cell-depleted fraction after *Sox5* overexpression ($P = 0.02$). *Sox5* overexpression (70- and 7.5-fold in two experiments) increased insulin secretion both in response to 16.7 mM glucose (40%, $P = 0.01$) and 70 mM $K^+$ (21%, $P = 0.03$; Supplementary Fig. 5h).

**Effects of *Sox5* knockdown and overexpression on T2D module.** SOX5 has putative binding sites to 123 of the 168 open chromatin genes (Supplementary Fig. 6a). *Sox5*-kd induced pronounced effects on the expression of the T2D-associated module in INS-1

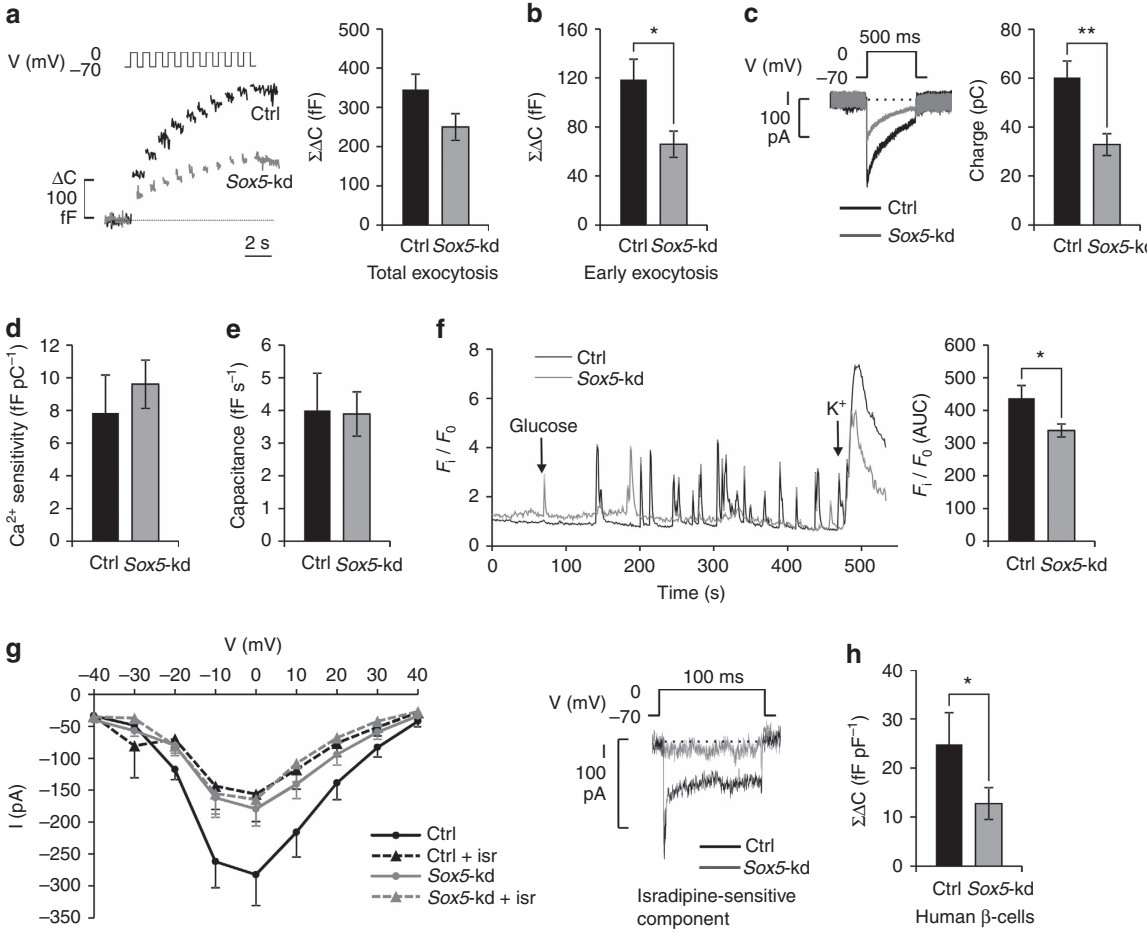

**Figure 4 | Effects of *Sox5*-kd on exocytosis and Ca$^{2+}$ currents.** The experiments were performed in INS-1 832/13 cells unless stated otherwise. *Sox5*-kd cells were transfected with Sox5 siRNA 48 h before the experiment. Control cells were transfected with negative control siRNA. (**a**) Increase in cell capacitance (ΔC), reflecting exocytosis, evoked by a train of ten 500 ms depolarizations from −70 to 0 mV applied to simulate glucose-induced electrical activity. Bar graph shows total capacitance increase (ΣΔC) in *Sox5*-kd and control (Ctrl) cells. Data from 21–25 cells per group. (**b**) Capacitance increase (ΣΔC) in response to the first two depolarizations of the train ('early exocytosis'). Data are from the same cells as in **a**. (**c**) The Ca$^{2+}$ current (I) in response to the first depolarization of the train. The bars denote the integrated Ca$^{2+}$-current (charge) in *Sox5*-kd and control cells. Data from the same cells as in **a**. (**d**) Average Ca$^{2+}$-sensitivity of the exocytotic process (defined as the capacitance increase divided by the integrated Ca$^{2+}$ current in response to the first depolarization) in *Sox5*-kd and control cells. Data from the same cells as in **a**. (**e**) Average capacitance increase in response to infusion of a solution containing high Ca$^{2+}$ concentration (free [Ca$^{2+}$]$_i$ ∼1.5 μM) ($n=10$–11). (**f**) Representative recordings of Fluo-5F fluorescence in *Sox5*-kd and control cells. Data are presented as the ratio of Fluo-5F fluorescence ($F_i$) normalized to the basal values at 2.8 mM glucose in control cells ($F_0$). The bar graph shows the area under the curve (AUC) of the fluorescence signal during stimulation with 20 mM glucose ($n=10$–25). (**g**) Current–voltage relationship for the peak Ca$^{2+}$ current in the absence or presence of 2 μM isradipine (isr) in *Sox5*-kd and control cells ($n=8$–15 cells). The isradipine-sensitive component was obtained by subtracting currents recorded in the presence of isradipine from currents observed in the absence of the blocker. (**h**) Total capacitance increase (ΣΔC) normalized for cell size in human β-cells treated with siRNA targeting *SOX5* (*SOX5*-kd) and control (Ctrl) cells ($n=6$–9 cells). One-sided *t*-test was used for statistical analysis. Data are mean ± s.e.m. *$P<0.05$; **$P<0.01$ using Student's *t*-test.

832/13 cells, as analysed by microarray. A total of 869 of the 2,889 module genes that were annotated on the array were differentially expressed in *Sox5*-kd cells relative to control cells (at nominal $P<0.05$), which represents a 1.5-fold enrichment over what would be expected by chance ($P<1$E-6 using Fisher's Exact test; $n=3$ microarrays from each condition). We confirmed this analysis in an independent experiment (1.5-fold enrichment; $P<1$E-6; $n=3$). Overexpression of *Sox5* (12-fold increase) affected 421 of the module genes (at nominal $P<0.05$), which also represents a significant enrichment (1.4-fold; $P<1$E-6; $n=3$). Notably, the gene expression changes in response to *Sox5*-kd were highly similar to those observed in islets from T2D donors relative to non-diabetic donors (same direction of change; Fig. 5a and Supplementary Table 9), whereas the opposite response was observed following *Sox5* overexpression

($P=0.003$; 1.9-fold enrichment of genes having a similar expression change to that in T2D following *Sox5*-kd and the opposite change after *Sox5* overexpression at $P<0.05$ for both *Sox5*-kd and *Sox5* overexpression). Genes exhibiting such consistent expression changes in response to *Sox5* perturbation had on average higher connectivity ($P=0.047$ using independent *t*-test) and were more strongly correlated with T2D status ($P=1$E-6 using independent *t*-test) compared with module genes that were unaffected by *Sox5*-kd and overexpression. They were also enriched for genes involved in membrane function ($P=0.04$) and ion channel activity ($P=0.03$). Several genes of importance to the Krebs cycle and the respiratory chain complex were downregulated by *Sox5*-kd (Supplementary Table 10), corroborating the metabolic data demonstrating impaired mitochondrial function.

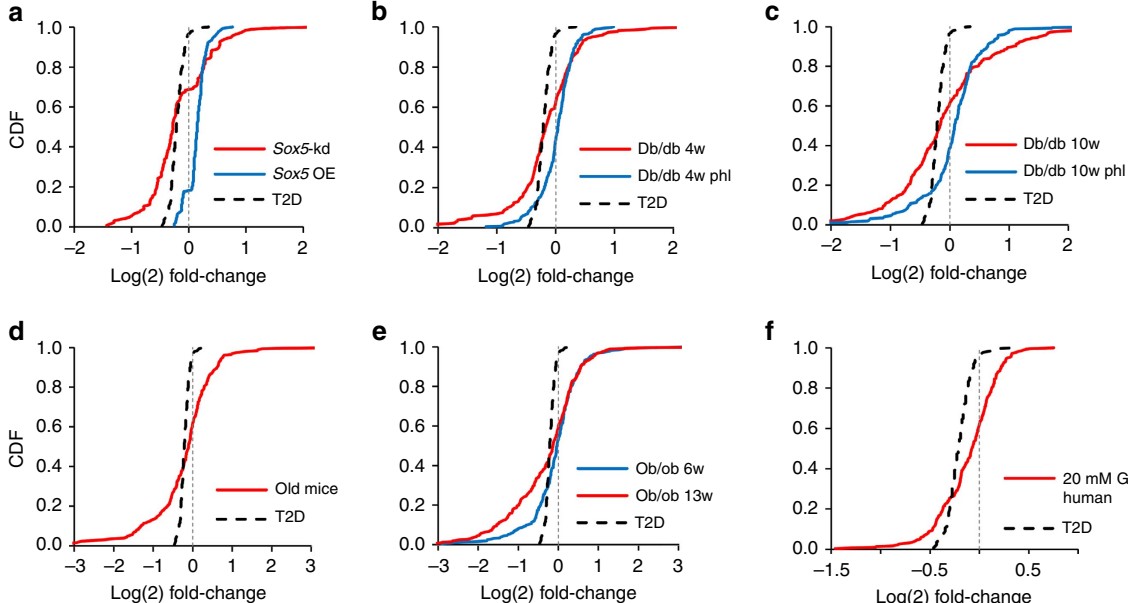

**Figure 5 | Expression of the T2D-associated module after *SOX5* perturbation and in animal models.** The cumulative density function (CDF) plot of the 168 open chromatin genes in human islets from T2D versus non-diabetic donors is displayed as a reference (dashed line) in all panels. (**a**) Cumulative density function (CDF) plots of log2-transformed expression fold-change of genes in the T2D-associated module in INS-1 832/13 cells transfected with *Sox5* siRNA (*Sox5*-kd; red) or a *Sox5* plasmid (green) relative to control cells ($n = 3$). (**b**) CDF plots of log2-transformed gene expression fold-change for the T2D-associated module in islets from 4-week-old db/db mice ($n = 3$) versus db/+ littermates ($n = 5$) and from phlorizin-treated db/db mice ($n = 3$) versus untreated db/db mice ($n = 5$). Phlorizin was administered at 400 mg kg$^{-1}$ daily for 10 days. (**c-f**) As in (**b**) but for 10-week-old db/db mice treated with phlorizin (400 mg kg$^{-1}$ daily) for 7 days ($n = 7$ untreated db/db and $n = 3$ phlorizin-treated db/db) (**c**), for 14-month-old C57BL/6 mice ($n = 6$) versus 8-week-old C57BL/6 mice ($n = 5$) (**d**), for 6 or 13-week-old ob/ob mice versus ob/+ littermates ($n = 5$ in each group) (**e**), and for human islets incubated at 20 mM glucose for 72 h versus islets incubated at 5.6 mM glucose ($n = 6$) (**f**).

***Sox5* and gene signature perturbation in animal models of T2D.** To further investigate the relevance of the T2D-associated module in β-cell failure, we analysed global gene expression in islets from animal models characterized by increased β-cell stress. In islets from db/db mice at 4 weeks of age (when they are still normoglycemic; non-fasting blood glucose 8.0 ± 0.3 mM), *Sox5* expression was reduced by 20% and a large fraction of the module genes were changed in a similar direction to that observed in T2D islets ($P < $1E-6 using Fisher test; 1.4-fold enrichment; Fig. 5b). The expression perturbations were even more pronounced in islets from db/db mice at 10 weeks of age. These mice had a 49% reduction of *Sox5* expression, and compared with the pattern at 4 weeks there was a 47 ± 19% higher differential expression of the signature genes relative to age-matched control islets ($P < $1E-6; Fisher test; Fig. 5c). At this age db/db mice have severely blunted insulin secretion[24] and overt hyperglycaemia (non-fasting blood glucose 19.0 ± 3 mM). The gene expression changes could be reversed (Fig. 5b,c) by treating the mice with phlorizin, which increases renal glucose excretion[25].

In islets from 14-month-old mice, *Sox5* expression was reduced by 80% and a significant fraction of the module genes were perturbed relative to young (8 weeks) mice (1.5-fold enrichment; $P < $1E-6; Fisher test; Fig. 5d). Old mice have been shown to have increased β-cell stress[17] despite being normoglycemic (non-fasting blood glucose 7.0 ± 0.4 mM).

Islets from 6-week-old ob/ob mice displayed no significant perturbation of the module (Fig. 5e). By contrast, in 13-week-old ob/ob islets, *Sox5* expression was reduced by 52% and a large fraction of the module genes were changed (1.3-fold enrichment; $P < $1E-6; Fisher test). At this age the ob/ob mice are normoglycemic but highly insulin-resistant with increased β-cell demands[24]. We want to stress that the comparison of mouse models to T2D in humans can only be correlated with changes in *Sox5* expression as many other genes and pathways are also affected in these animal models.

Culturing human islets at 20 mM glucose for 3 days did not significantly affect the module genes (Fig. 5f). It is, however, not surprising that culturing human islets in high glucose for such a short time period does not correlate with the T2D module, since T2D occurs over decades and involves multiple organ systems. Accordingly, we observed no consistent expression changes of the signature genes when interrogating a data set of human islets exposed to 48 h palmitate treatment[26].

Pancreatic β-cells from both db/db mice and old mice have been shown to exhibit dedifferentiation, as the expression of key transcription factors is reduced and genes associated with immature cell states are being expressed[17,25]. Our data, therefore, suggest that the gene expression signature in human islets from T2D donors is reminiscent of a dedifferentiation profile. The signature is not merely secondary to hyperglycemia, since it was perturbed in several models characterized by normoglycemia but increased β-cells stress and was unaffected by culturing islets at high glucose. We want to emphasize that all comparisons between T2D islets and animal models and short-term culture should be interpreted with caution as T2D occurs over decades and involves multiple organ systems.

**Regulation of *Sox5* expression.** The upstream regulation of *Sox5* expression remains unclear. *Sox5* expression in INS-1 832/13 cells was reduced both in response to 20 mM glucose and 0.5 or 1 mM palmitate (Supplementary Fig. 6b).

As Foxo1 has recently been suggested to be involved in β-cell dedifferentiation in animal models[17] we analysed the effects of

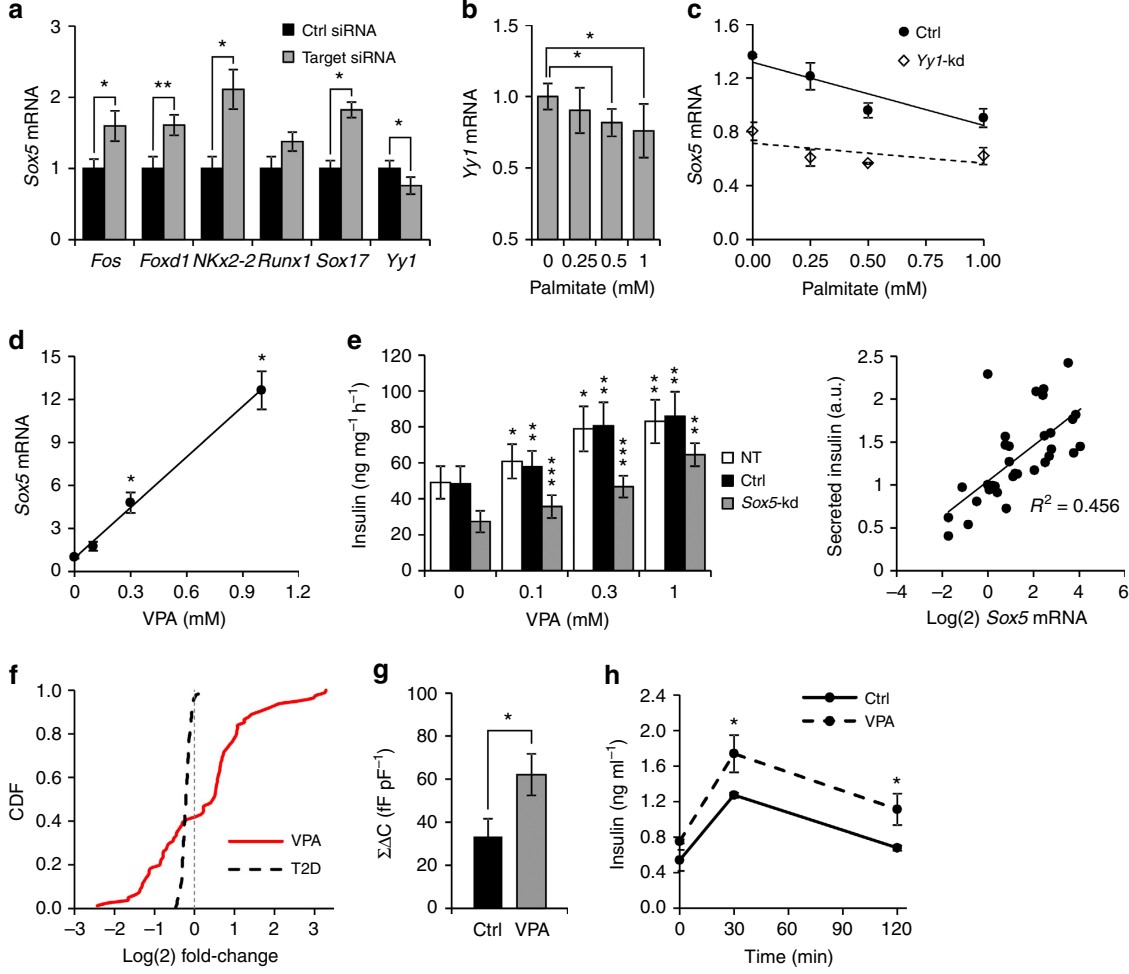

**Figure 6 | Characterization of the regulation of Sox5.** The experiments were performed in INS-1 832/13 cells if not stated otherwise. *Sox5*-kd cells were transfected with Sox5 siRNA 48 h before the experiment, and control cells were transfected with negative control siRNA. (**a**) *Sox5* mRNA levels relative to control (Ctrl) cells after knockdown of putative regulators of *Sox5* expression ($n = 4$). (**b**) *Yy1* mRNA levels after 48 h incubation with different concentrations of palmitate ($n = 3$). (**c**) Effect of palmitate on *Sox5* expression in control cells and after knockdown of *Yy1* (*Yy1*-kd) ($n = 3$). (**d**) *Sox5* mRNA levels after 48 h treatment with different concentrations of valproic acid (VPA) relative to untreated cells ($n = 3$). (**e**) Insulin secretion at 16.7 mM glucose after incubation with or without VPA for 48 h in non-transfected cells (NT), cells transfected with a negative control siRNA (Ctrl) and *Sox5*-kd cells ($n = 4$ per condition). Insulin secretion was plotted against log2-transformed expression levels of *Sox5* for each of the 12 conditions indicated in the bar graph, with values from three independent experiments. (**f**) Cumulative density function (CDF) plot of log2-transformed expression fold-change of genes in the T2D-associated module in cells treated with 1 mM VPA relative to control cells. (**g**) Total capacitance increase ($\Sigma\Delta C$) in cells treated with 0.3 mM VPA for 48 h relative to untreated cells ($n = 12$–14 cells). (**h**) Serum insulin concentration at 0, 30 and 120 min during an intraperitoneal glucose tolerance test in 12-week-old female NMRI mice treated with 250 mg kg$^{-1}$ VPA or vehicle (Ctrl) daily for 7 days ($n = 3$ mice per condition). Insulin levels were increased by 37% at 30 min and by 64% at 120 min of the IPGTT ($P = 0.046$ and 0.037, respectively. All mice were normoglycemic and there was no difference in glucose tolerance between the groups. Data are mean ± s.e.m. *$P < 0.05$; **$P < 0.01$; ***$P < 0.001$. Student's *t*-test was used in **a**,**b**,**c**,**d**,**g**,**h**, and Pearson correlation test was used in **e**.

*Foxo1* silencing on *Sox5* expression. However, *Foxo1* silencing (by 74 ± 7%) did not affect *Sox5* expression or glucose-stimulated insulin secretion (Supplementary Fig. 6c).

We next analysed transcriptional binding sites in the vicinity of *Sox5* and identified 39 putative regulators. We conducted a siRNA screening to silence each of the 39 genes in INS-1 832/13 cells (Supplementary Fig. 6d) and observed significant changes in *Sox5* expression after knockdown of five of them (Fig. 6a). *Sox5* expression was increased in response to silencing of both *Nkx2.2* and *Sox17*, which are important for pancreatic development. It is tempting to speculate that these transcription factors might regulate *Sox5* expression during islet development, although the role of *Sox5* in islet development remains to be studied. In contrast, *Sox5* expression was reduced after silencing the

transcription factor Yin Yang 1 (*Yy1*), which was in turn significantly affected by palmitate treatment (Fig. 6b). *Yy1* regulates genes involved in cellular differentiation, mitochondrial function and stress response and has been suggested to protect against oxidative stress in β-cells[27]. There was a strong association between *Yy1* and *Sox5* expression ($P = 5E-7$; linear regression; Supplementary Fig. 6e). Moreover, the inhibitory effect of palmitate on *Sox5* expression was abolished after *Yy1* knockdown (Fig. 6c). *Yy1* expression was also associated with glucose-stimulated insulin secretion ($P = 0.004$; linear regression; Supplementary Fig. 6f). This association was however non-significant when correcting for *Sox5* expression ($P = 0.93$), while the association between *Sox5* expression and insulin secretion was significant ($P = 0.035$; linear regression;

Supplementary Fig. 6g) even after correcting for *Yy1* expression. Taken together the findings point to a vicious cycle in which elevated nutrients, partly via *Yy1*, decrease *Sox5* expression and lead to impaired insulin secretion, which in turn may aggravate the hyperglycemia.

**Valproic acid elevates *Sox5* expression and insulin secretion.** On the basis of the present findings, we postulate that the 168 open chromatin genes represent a core set of highly connected genes in the T2D-associated module that are suppressed in T2D, resulting in loss of secretory function. We therefore wanted to investigate the effect on gene expression and β-cell function of the HDAC inhibitor valproic acid (VPA) that remodels chromatin structure.

Incubation of INS-1 832/13 cells with VPA at 0.1, 0.3 or 1 mM for 48 h significantly increased *Sox5* expression (Fig. 6d). Moreover, VPA dose-dependently increased insulin secretion (Fig. 6e). Silencing *Sox5* attenuated but did not fully abolish the stimulatory effect of VPA on insulin secretion, which might be a result of incomplete knockdown of *Sox5* and the involvement of additional genes to the effect of VPA (Supplementary Fig. 7a). In islets from C57BL/6 mice, 1 mM VPA elevated *Sox5* mRNA fourfold and increased insulin secretion in response to 16.7 mM glucose (7.5-fold; $P = 0.02$) and 70 mM K$^+$ (6.4-fold; $P = 0.01$). The response to VPA was further potentiated by 72 h co-incubation with palmitate (Supplementary Fig. 7b).

We observed a clear association between *Sox5* expression and glucose-stimulated insulin secretion at all VPA doses (Fig. 6e). To formally assess whether the increased *Sox5* expression in response to VPA is causally associated with improved insulin secretion or rather represents a reactive or independent change, we applied a causal inference test (CIT)[28] based on four prerequisites. First, VPA dose and *Sox5* expression were associated ($P < 1E-6$ by linear regression; $\beta = 10.5$). Second, VPA dose and insulin secretion were associated ($P = 0.0001$; $\beta = 0.73$). Third, VPA dose and *Sox5* expression were associated conditional on insulin secretion capacity ($P < 1E-6$; $\beta = 9.7$). Fourth, there was no association between the VPA dose and insulin secretion when data were adjusted for *Sox5* expression ($P = 0.3$). The omnibus $P$ value of the test suggests a causal relationship between VPA dose, *Sox5* expression and insulin secretion ($P = 0.03$ for causal relationship, $P = 0.75$ for reactive) implying that the stimulatory effect of VPA on insulin secretion is mediated by enhanced *Sox5* expression.

Intriguingly, the majority (61%) of the genes in the T2D module that were differentially expressed following *Sox5* overexpression (at nominal $P < 0.05$) were also significantly affected by VPA ($P < 1E-6$ for the enrichment; Fisher test). Of the 168 open chromatin genes, 58 were differentially expressed between cells exposed to *Sox5*-kd and *Sox5* overexpression (at $P < 0.05$), while VPA changed 81 of the 168 genes (41 in common, 1.6-fold enrichment; $P < 1E-6$; Fisher test; Supplementary Table 11). Taken together, the present data show that VPA, similar to *Sox5* overexpression, affects a significant fraction of the genes in the T2D signature in human islets (Fig. 6f) and demonstrate that VPA enhances insulin secretion via increased *Sox5* expression. It is important to emphasize that VPA has global effects on gene expression (1,348 other module genes were significantly affected) and is therefore not specific to *Sox5* despite the association between *Sox5* expression and VPA dose.

We also observed that VPA increased exocytosis by 86% in INS-1 832/13 cells (Fig. 6g). Moreover, we treated NMRI mice with VPA for 7 days and observed increased insulin secretion *in vivo* by an intraperitoneal (i.p.) glucose tolerance test (Fig. 6h).

**SOX5 overexpression increases human islet insulin secretion.** Finally, we investigated the effects of *SOX5* knockdown or overexpression in human cells. In the human β-cell line EndoC-BH1 (ref. 29), *SOX5* knockdown (74 ± 1% silencing) reduced glucose-stimulated insulin secretion by 29% ($P = 0.004$; Fig. 7a). We next overexpressed *SOX5* (2.8 ± 0.4-fold; Supplementary Fig. 8a) in human islets, which increased insulin secretion both in response to 16.7 mM glucose ($P = 0.01$; Fig. 7b) and 70 mM K$^+$ ($P = 0.01$). There was no difference in insulin content between control islets and islets with *SOX5* overexpression (Supplementary Fig. 8b). We hypothesized that the effect of *SOX5* overexpression would be greater in T2D islets and therefore analysed glucose-stimulated insulin secretion separately in non-diabetic versus T2D islets (Fig. 7c). In non-diabetic islets insulin secretion in response to 16.7 mM glucose was 4.3% (of total insulin content) compared with 3.3% in T2D islets ($P = 0.04$). Overexpression of *SOX5* in non-diabetic islets increased glucose-stimulated insulin secretion by 0.23 percentage points (of insulin content), while secretion was improved by 0.68 percentage points in T2D islets ($P = 0.04$), partly restoring the impaired insulin release.

We also analysed 35 common genetic risk variants for T2D but found no association with *SOX5* expression or the eigengene representing the 168 open chromatin genes in human islets (Supplementary Table 12).

We have previously identified a genetic risk score (0–8 risk alleles) that is associated with reduced insulin secretion in human islets[5]. There was no correlation between the risk score and *SOX5* expression. However, at four or more risk alleles, the effect of *SOX5* expression on insulin secretion was significantly more pronounced ($\beta = 1.26$; $P = 0.02$; linear regression) compared with islets from donors with three or fewer risk alleles ($\beta = 0.17$; $P = 0.08$; $P = 0.001$ for comparison of the beta values using $t$-test). The findings suggest that the perturbations of *SOX5* and the T2D-associated module are not directly caused by common genetic risk variants (but rather by β-cell stress). However, the manifestation of these changes appears to be influenced by DNA variants, such that genetically susceptible individuals develop a more severe secretory failure in response to β-cell stress. This model would be in agreement with the small effect size of each of the individual genetic risk variants identified for T2D.

We finally analysed eight key genes that are well-known markers of differentiated human β-cells. These genes were all downregulated in human islets from T2D donors compared with non-diabetic donors (Supplementary Table 13), in agreement with a dedifferentiation pattern. We also observed that six of these eight genes were significantly upregulated after SOX5 overexpression in human T2D islets (Fig. 7d). Moreover, *SOX5* overexpression increased the expression of *CACNA1D* mRNA ($P = 0.007$; Fig. 7d).

Taken together, the human data corroborate the observations in INS-832/13 cells and rodent islets, showing that *SOX5* overexpression induces expression changes opposite to that of T2D islets, increases *CACNA1D* expression and improves β-cell function in T2D islets.

**Discussion**

The present data identify a T2D-associated co-expression module that is enriched for genes with islet-selective open chromatin. The expression pattern of these genes in human islets from T2D donors is highly reminiscent to that of dedifferentiated β-cells. We also identify *SOX5* as a regulator of the module. In addition to *SOX5*, we also observed a consistent downregulation of *LPAR1*, *PTCH1*, *SMARCA1*, *TCF3*, *TMEM196* and *TMEM63C* both in

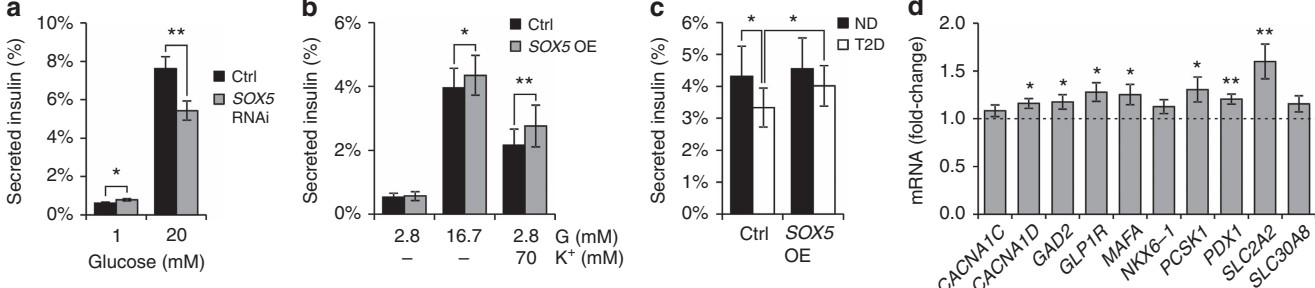

**Figure 7 | Effects of *SOX5* knockdown in a human β cell line and of *SOX5* overexpression in human islets. (a)** Insulin secretion (as per cent of total insulin content) from EndoC-BH1 cells during a 1-h incubation with 1 or 20 mM glucose after lentiviral RNAi-mediated *SOX5* knockdown ($n = 3$). **(b)** Insulin secretion (as % of total insulin content) from batch-incubated human islets after *SOX5* lentiviral overexpression relative to islets transduced with control lentivirus. Islets were incubated for 1 h with glucose (G) and for 15 min with $K^+$ as indicated ($n = 3$–9 replicates from eight donors per condition, of which three had T2D). **(c)** Insulin secretion (as % of total insulin content) during a 1-h incubation with 16.7 mM glucose in non-diabetic and T2D islets with or without *SOX5* overexpression, respectively ($n = 4$–9 replicates from five non-diabetic donors and three T2D donors). **(d)** mRNA expression of genes in human islets after *SOX5* overexpression relative to control islets ($n = 10$ donors). Log2-transformed gene expression data were used for statistical analysis (paired *t*-test). Data are mean ± s.e.m. *$P < 0.05$; **$P < 0.01$ using Student's *t*-test.

response to high palmitate and glucose. We cannot exclude that these other genes may also affect islet function and the gene co-expression module as there are often many different regulators that act in concert.

Insulin secretion following *Sox5*-kd was compromised by impaired mitochondrial activity, reduction of L-type $Ca^{2+}$ channel expression and decreased depolarization-evoked $Ca^{2+}$ influx. Intracellular $Ca^{2+}$ regulates the activity of several enzymes, including ATP synthase[30]. The mitochondrial defect and the impaired $Ca^{2+}$ influx could therefore act in concert to aggravate the secretory defect.

Human T2D develops through a vicious cycle, characterized by progressive changes in a plethora of genes leading to metabolic perturbations, including β-cell dysfunction[18,22]. Chronically elevated nutrient intake increases the secretory demands of the β-cell, which produces a compensatory response that initially maintains euglycemia but also evokes β-cell stress[18]. In this context it is pertinent that db/db islets from 4-week-old mice, which are normoglycemic but have increased insulin secretory demands[24], display expression perturbations similar to those of T2D human islets. These gene expression changes were reversed by treatment with phlorizin, which could be assumed to alleviate the β-cell demands[25].

In contrast to db/db mice, ob/ob mice compensate for the increased insulin requirements over long time. However, ob/ob islets display declining secretory response with time, likely due to protracted β-cell stress[24]. Accordingly, we found that the gene signature was unaffected in islets from 6-week-old ob/ob mice but was perturbed at 13 weeks. It is also of interest that both *SOX5* ($P = 0.02$) and the T2D-associated gene signature ($P = 0.05$) had reduced expression in human islets from non-diabetic donors ($n = 45$) with BMI above median ($27 \, \mathrm{kg \, m^{-2}}$), which may be at higher risk of developing T2D, compared with islets from non-diabetic donors with BMI below the median.

On the basis of the present data we propose a model for T2D in which decreased *Sox5* expression contributes to reduced expression of genes with islet-selective open chromatin and loss of β-cell secretory function, due to both metabolic and distal secretory defects involving reduced $Ca^{2+}$ influx. The expression perturbations are similar to those observed in models exhibiting β-cell dedifferentiation, although the altered β-cell state in the pathophysiology of human T2D may not be as severe as in some genetic models[17,31]. There was no induction of *NGN3*, *NANOG* or other developmental progenitor markers in the T2D islets, and

the T2D signature may therefore also be described as 'immaturity' or 'loss of β-cell identity'. Our data suggest that changes of *SOX5* and the gene module represent early events in the vicious cycle that leads to β-cell failure which is precipitated by chronically elevated nutrient intake (for example, glucotoxicity). Our data also show that impaired glucose-stimulated insulin secretion in T2D islets can be restored by *SOX5* overexpression. *SOX5* overexpression improved insulin secretion by 18% in T2D islets (secretion increased from 3.4 to 4.0% of total insulin content) but had a mere 6% effect in non-diabetic islets (secretion increased from 4.3 to 4.5% of insulin content). Secretory function was not completely restored by *SOX5* overexpression, which is not surprising considering that a plethora of genes are perturbed in T2D and islet dysfunction may be partly irreversible after long-standing disease.

The central role of the islet-selective open chromatin genes is shown by the striking effects of VPA on β-cell gene expression and function. VPA has not been clinically investigated for T2D but has been associated with hyperinsulinemia in patients with epilepsy who receive the drug[32] and has been shown to stimulate insulin secretion *in vitro*[33]. Our new findings that VPA improves insulin secretion via elevated *Sox5* expression, restores the T2D-associated module and increases glucose-stimulated insulin secretion *in vivo* raise the exciting possibility that VPA and other HDAC inhibitors could be a potential means to treat defective insulin secretion in T2D by counteracting an immature β-cell state.

## Methods

**Human islets.** Experimental procedures were approved for Lund University by the Regional Ethical Review Board in Lund. Donated human islets were obtained (with research consent) from the Nordic Network for Clinical Islet Transplantations (Professor O. Korsgren). Some of these islets, but not the full cohort used here, have been utilized in other studies from Lund University Diabetes Centre[5,11,34]. Islets were extracted from multi-organ donors[35,36]. The pancreas was perfused with ice-cold collagenase, cut into pieces and placed in a digestion chamber at 37 °C. Separation of endocrine and exocrine tissues was achieved by a continuous density gradient. Selected fractions were then centrifuged to enrich for islets. Purity of islets was measured by dithizone staining[35]. From this suspension, islets to be used for experiments were hand-picked under a microscope. The islets were cultured at 5.6 mM glucose in CMRL-1066 (INC Biomedicals) supplemented with 10 mM HEPES, 2 mM L-glutamine, 50 µg ml$^{-1}$ gentamicin, 0.25 µg ml$^{-1}$ Fungizone (GIBCO), 20 µg ml$^{-1}$ ciprofloxacin (Bayer Healthcare), 10 mM nicotinamide and 10% human serum at 37 °C (95% $O_2$ and 5% $CO_2$) for 1–9 days before experiments. Donors with known T2D or HbA1c > 6.0% and no GAD antibodies were defined as having T2D. There was no difference in islet purity

between non-diabetic and T2D donors. See Supplementary Tables 1 and 5 for data on donors.

**Animals.** Male db/db and db/+ mice (4 and 10 weeks of age) were from Charles River, and male ob/ob and ob/+ mice (6 and 13 weeks of age) were from Janvier labs. A 14-months-old male C57BL/6J mice (kept at the local animal facility for 12 months) and 12-week-old female NMRI mice were from Taconic. All animal experiments were approved for Lund University by the ethical committee for animal research in Malmö/Lund.

**Rat and mouse islets.** Pancreatic islets from rats or mice were prepared by collagenase digestion of the exocrine pancreas. The islets were hand-picked in Hank's buffer (Sigma-Aldrich) with 1 mg ml$^{-1}$ BSA and either used directly for mRNA extraction or incubated in a humidified atmosphere in RPMI 1,640 tissue culture medium containing 5 mM glucose (SVA, Sweden) supplemented with 10% (vol/vol) fetal bovine serum, 100 IU ml$^{-1}$ penicillin and 100 µg ml$^{-1}$ streptomycin. RNA from mouse islets used for microarray analysis was prepared with the miRNeasy kit (Qiagen).

**VPA treatment.** NMRI mice were injected i.p. with VPA (250 mg kg$^{-1}$ daily) or vehicle (PBS) for 7 days before performing an i.p. glucose tolerance test.

**Phlorizin treatment of db/db mice.** Phlorizin (Sigma-Aldrich) was prepared as a 20% stock (0.2 g ml$^{-1}$) in 1,2-propanediol (Sigma-Aldrich) and kept at 4 °C. Mice were injected subcutaneous with 0.4 g kg$^{-1}$ phlorizin or vehicle once daily for 7 days (10-week-old db/db; 3 phlorozin-treated and 7 vehicle-treated mice) or 10 days (4-week-old db/db; 3 phlorozin-treated and 7 vehicle-treated mice). Non-fasted blood glucose was measured before killing. For the phlorizin-treated 4-week-old db/db mice it was 4.7 ± 0.4 mM and for the control-treated 4-week-old db/db mice it was 4.8 ± 0.6 mM. For the phlorizin-treated 10-week-old db/db mice it was 8.4 ± 1.5 mM and for the control-treated 10-week-old db/db mice it was 20.0 ± 3.2 mM. Animals were killed by cervical dislocation and pancreatic islets and RNA was prepared as described above.

**INS-1 832/13 cells.** Rat insulinoma INS-1 832/13 cells developed by Hohmeier et al.[37] and kindly provided by Dr. Hindrik Mulder were used for experiments involving siRNA and Sox5 overexpression. INS-1 832/13 cells (passages 55–70) were cultured at 10 mM glucose in RPMI 1640 (Life Technologies) and supplemented with 10% (vol/vol) fetal bovine serum, 100 IU ml$^{-1}$ penicillin, 100 µg ml$^{-1}$ streptomycin, 10 mM HEPES, 2 mM glutamine, 1 mM sodium pyruvate, and 50 µM β-mercaptoethanol.

**EndoC BH1 cells.** Human EndoC-BH1 cells, kindly provided by Dr Raphael Scharfmann, were used for lentiviral RNA interference experiments. EndoC-BH1 cells were cultured in DMEM low glucose (1 g l$^{-1}$, Thermo Fisher Scientific), 2% albumin from bovine serum fraction V, 50 µM β-mercaptoethanol, 10 mM nicotinamide, 5.5 µg ml$^{-1}$ transferrin, 6.7 ng ml$^{-1}$ sodium selenite, 100 IU ml$^{-1}$ penicillin and 100 µg ml$^{-1}$ streptomycin. The culture vials were pre-coated with DMEM (4.5 g l$^{-1}$ glucose) containing 100 IU ml$^{-1}$ penicillin, 100 µg ml$^{-1}$ streptomycin, 2 µg ml$^{-1}$ fibronectin (Sigma-Aldrich) and 1% ECM (Sigma-Aldrich) for at least 4 h. For glucose starvation before secretion experiments, glucose-free DMEM (Thermo Fisher Scientific) was used as a base and glucose added to 2.8 mM.

**Microarray analysis of islets and INS-1 832/13 cells.** Total RNA was extracted from human islets with the AllPrep DNA/RNA Mini Kit (Qiagen). RNA quality and concentration were measured using an Agilent 2100 Bioanalyzer (Bio-Rad) and a Nanodrop ND-1000 (NanoDrop Technologies). The Affymetrix GeneChip Human Gene 1.0 ST microarray chip (Affymetrix) was used for gene expression analysis in accordance with the standard protocol. Briefly, total RNA was converted into biotin-targeted cDNA following the manufacturer's specifications, and the biotin-labelled cDNA was fragmented into strands with 35–200 nucleotides. This was hybridized onto the chip overnight in a GeneChip Hybridization 6400 oven using standard procedures. The arrays were washed and stained in a GeneChip Fluidics Station 450. Scanning was carried out with the GeneChip Scanner 3000 and image analysis was done with the GeneChip Operating Software. Data normalization was performed using Robust Multi-array Average.

Microarray analysis of INS-1 832/13 cells treated with siRNA, Sox5 plasmid or VPA, was performed using SurePrint G3 Rat GE 8x60K V2 Microarray Kit. Microarray analysis of islets from db/db,db/+, ob/ob, ob/+, C57BL/6J and NMRI mice was done using SurePrint G3 Mouse GE 8 × 60K V2 Microarray Kit. Microarray analysis of human islets cultured at low or high glucose was done using SurePrint G3 Human GE 8 × 60K V2 Microarray Kit. Data were pre-processed using quantile normalization in Partek Genomic Suite.

**Co-expression network analysis.** The co-expression network analysis was performed in R (version 2.15.1) using log2-transformed microarray expression data from human islets. Using the WGCNA framework[12] and the corresponding Bioconductor package[38], we first calculated the pair-wise co-expression for all genes and formed a similarity matrix based on the Pearson correlation coefficients $s_{i,j} = |cor(x_i, x_j)|$, where $x_i$ denotes the expression vector for gene i across the samples.

Next, the similarity matrix was transformed into an adjacency matrix $a_{i,j} = |cor(x_i, x_j)|^\beta$. The connectivity of a gene in a network (the degree k) equals the sum of all connections for that gene.

Biological networks have been suggested to exhibit a scale-free property[9], which means that the probability that a node is connected with k other nodes (the degree distribution p(k)) decays as a power function $p(k) \sim k^{-\gamma}$. Linear regression analysis of log-transformed k and p(k) was used to estimate how well the co-expression network satisfied the scale-free topology for different values of β. We found that for $\beta \geq 5$ ($\beta \geq 8$ in the replication set) $R^2$ for the fit was >0.8. On the basis of the adjacency matrix, the topological overlap, which reflects the relative gene interconnectedness, was calculated for all gene pairs[9]. The non-negative and symmetric topological overlap matrix $\Omega = [\omega_{i,j}]$ was converted to dissimilarity (distance) measures $d_{i,j} = 1 - \omega_{i,j}$, which were used for module identification. The eigengene, defined as the 1st principal component of the gene expression matrix, was determined for each module and was correlated with the phenotype traits. The human islet insulin secretion data showed non-Gaussian distributions. Gaussian distribution was obtained using logarithm transformation, and analyses of human islet insulin secretion data were therefore performed using log-transformed data.

For each gene in the T2D-associated module the connectivity within the module ($k_{in}$) was determined. In addition, the correlation between the gene expression trait and T2D status was calculated. Next, the correlation between gene connectivity and the trait associations was analysed across the genes by Spearman's rank correlation.

**Comparisons with public microarray data.** Gene expression profiles from >8,100 publically available microarray data sets (www.ncbi.nlm.nih.gov/geo and https://www.ebi.ac.uk/arrayexpress/) analysed by Affymetrix, Agilent or Illumina chips were downloaded and processed (all Affymetrix, Agilent or Illumina chips from which raw data were available, including all tissue and conditions). Only data sets for which the full raw data were available were included. The processed data are freely available at www.trialomics.com. The probe-level data were processed using the Supervised Normalization of Microarrays (SNM) framework to normalize for array effects, detect and remove outlier arrays, and facilitate cross-data set analysis. Our specific implementation of this framework were as follows: (1) for each array, we calculated the average value of all probes aligning to gene g; (2) we defined an overall average for gene g across all microarrays from that platform; (3) for every sample, we used a b-spline basis function to remove any intensity-dependent variation between the gene-level mean values calculated in step 1 with the average values calculated in step 2 as the reference.

Next, summary statistics were calculated for each gene in each study. These statistics were then aggregated into matrices where each row corresponds to a gene, each column a study, and each element a summary statistic (mean or variance). The corresponding matrix of gene variances across samples used to identify studies in which the 168-gene signature was, as a group, more variable than expected by chance. When comparing differential expression we used a chi-squared test to assess relationships between differentially expressed genes between studies.

**In silico analysis of TFBS.** TFBS were detected using the Regulatory Genomics Toolbox (regen.googlecode.com). A region 1 kilobases (kb) upstream of the gene promoter (Ensembl built GRCh37.p13) was used to find binding sites. Next, a motif match analysis was performed using the motif matching tool available in Biopython[39] with a false positive rate of 0.0001 in these regions[40]. Motifs were obtained in Jaspar, Transfact (public) and the Uniprobe databases[41,42]. The same procedure was repeated 100 times on random genomic regions with the same length as the original regions. We employed a one-tailed Fisher's Exact test to measure if the proportion of binding sites of a motif inside the gene promoters was higher than the proportion of binding sites in random regions. P values were corrected by the Benjamini–Hochberg method. For detection of candidate regulators of SOX5, we performed motif matches in open chromatin regions close to SOX5 (that is, overlapped with the gene or promoter region or 1 kb upstream).

**Genotype analyses.** Genomic DNA was extracted from human islets from all donors. The SNPs shown in Table S10 were genotyped using an allelic discrimination assay-by-design method on an ABI 7900 analyzer (Applied Biosystems). For global analysis of SNPs affecting the module eigengene, we used the Affymetrix Human SNP array 6.0. The association between the SNPs and SOX5 expression or the module eigengene was analysed by linear models and in-house R and Perl code (at Sage Bionetworks)[43]. For the analyses of expression SNPs (eSNPs), we made ten permutations of the expression file. The permutation was done by swapping columns preserving the co-expression structure of the data.

The analysis of SNP enrichment among the 168 open chromatin genes used the following procedure. (1) SNPs within 50 kb of each of the 168 genes were identified. (2) Published genome-wide association data were interrogated to identify SNPs

that were associated with insulin secretion measures (HOMA-B and corrected insulin response [CIR]). (3) SNPs associated with HOMA-B or CIR at $P < 0.001$ were used for further analyses. (4) The enrichment of SNPs in the vicinity of the 168 genes was assessed by SNP set enrichment analysis[44].

**Causal inference test.** The CIT[28] was used to assess whether there was a causal relationship between VPA dose, *Sox5* expression and glucose-stimulated insulin secretion such that VPA→*Sox5*→insulin secretion, or whether there was rather a reactive or independent relationship. The CIT takes into account all four prerequisites that have been suggested for inferring a causal relationship. VPA dose was treated as a categorical variable with no VPA, low dose (0.1 mM) or high dose VPA (1 mM). *Sox5* expression and insulin secretion were treated as continuous variables. The CIT determines an omnibus $P$ value for the causal relationship.

**Other bioinformatics analyses.** Gene enrichment analysis was performed using DAVID version 6.7 (ref. 45). Islet-selective open chromatin genes were identified as having open chromatin in the transcription starting site or gene body (Supplementary Table 2 in ref. 13).

**Transmission electron microscopy.** Human islets were fixated in 2.5% Glutaraldehyde in freshly prepared Millionig's buffer (1.88% $NaH_2PO_4 \cdot H_2O$ (Sigma-Aldrich), 0.43% NaOH, pH 7.2) and refrigerated for 2 h. After a wash in Millionig's buffer, the islets were post-fixated in osmium tetroxide (1%) for 1 h, and then carefully washed in Millionig's buffer. Finally, the islets were dehydrated and embedded in AGAR 100 (Oxford Instruments Nordiska AB, Sweden). Samples were cut into 70–90 nm ultrathin sections. The sections were placed on Cu-grids and contrasted with uranyl acetate and lead citrate before examination in a JEM 1,230 electron microscope (JEOL-USA. Inc., USA). Granules (large dense core vesicles) were defined as docked when the centre of the granule was located within 150 nm from the plasma membrane. The number of granule profiles within 150 nm from the plasma membrane was calculated using an in-house software programmed in MatLab 7 (MathWorks, Natick, USA)[46]. Electron micrographs were analysed from at least three different cells per donor (median seven cells).

**Insulin secretion and insulin content measurements in INS-1 832/13 cells.** Approximately 350,000–400,000 INS-1 832/13 cells were seeded per well in a 24-well plate 72 h before secretion experiments. Cells were washed twice with a secretion assay buffer (SAB) containing 114 mM NaCl, 4.7 mM KCl, 1.2 mM $KH_2PO_4$, 1.16 mM $MgSO_4$, 20 mM HEPES pH 7.2, 25.5 mM $NaHCO_3$, 2.5 mM $CaCl_2$, 0.2% BSA with 2.8 mM glucose. This was followed by 2 h preincubation at 37 °C with 5% $CO_2$ in 2 ml SAB. Next, the buffer was removed and cells were incubated for 1 h in SAB supplemented with glucose, tolbutamide (Sigma-Aldrich), clonidine (Sigma-Aldrich), diazoxide (Sigma-Aldrich), KCl (Sigma-Aldrich), carbachol (Sigma-Aldrich), L-Leucine (Sigma-Aldrich), α-ketosicaproic acid (Sigma-Aldrich) or Glp-1 (Bachem, Switzerland) as indicated. Immediately after incubation, an aliquot of the medium was removed and insulin was analysed directly or after storage at −20 °C for later analysis with the Coat-a-Count kit (Siemens) according to the manufacturer's protocol. For protein and total insulin analysis the remaining cells were lysed with RIPA buffer (50 mM Tris HCl pH 8, 150 mM NaCl, 1% NP-40/Triton X, 0.1% SDS, 0.5% sodium deoxycholate, 2 mM EDTA and 50 mM NaF). Cells were shaken on ice for 30 min followed by collection of the lysate and centrifugation at 10,000g for 5 min (4 °C). The supernatant was collected and analysed for total protein and insulin content directly or stored at −20 °C for later analysis. Total protein was measured with the Pierce BCA Protein Assay Kit (Thermo Scientific). Total insulin was assessed at a 1:100 dilution in PBS with the Coat-a-Count kit.

Average values based on technical replicates (2–8 wells) from three or more experiments were compared using Student's $t$-test. For the correlation analysis of insulin secretion versus mRNA expression (Fig. 6e), all secretion data were first normalized to values for non-treated cells.

**Insulin secretion and insulin content measurements in EndoC-BH1 cells.** EndoC-BH1 cells were seeded in 48-well plates at 100,000 cells per well 144 h before secretion experiments. The cells were cultured in medium containing 5.6 mM glucose, but 18–22 h before the experiments the medium was changed to medium containing 2.8 mM glucose. Cells were washed twice with secretion assay buffer (SAB; see above) containing 1 mM glucose and then preincubated in this buffer for 2 h at 37 °C with 5% $CO_2$. The buffer was removed and cells were incubated for 1 h in SAB supplemented with 1 or 20 mM glucose. Immediately after incubation, an aliquot of the medium was removed for measurement of secreted insulin, and the cells were lysed with RIPA buffer (see above) for analysis of total insulin content. Insulin levels were analysed using Mercodia Insulin ELISA.

**Insulin secretion and content measurements in human and mouse islets.** For static insulin secretion measurements, 10 or 12 islets were distributed into separate tubes and preincubated in 1 ml KREBS buffer (120 mM NaCl, 4.7 mM KCl, 2.5 mM $CaCl_2$, 1.2 mM $KH_2PO_4$, 1.2 mM $MgSO_4$, 10 mM HEPES, 25 mM $NaHCO_3$) with 2.8 mM glucose at 37 °C in a water bath for 30 min. The buffer was then exchanged

to KREBS buffer with 2.8 mM (with or without 70 mM $K^+$) or 16.7 mM glucose, and after 1 h incubation samples (800 µl) were collected and insulin content was analysed with the Millipore rat or human specific RIA kit. Some experimental series on human islets were analysed using Mercodia Insulin ELISA. The islets from each tube were collected and lysed in RIPA buffer for determination of total insulin and protein content.

**Incubation of rat islets and INS-1 832/13 cells with palmitate and glucose.** INS-1 832/13 cells were incubated for 48 h at 20 mM glucose or with 0.25, 0.5 and 1 mM palmitate. Palmitate-BSA stock solution (5 mM palmitate and 10% BSA)[47] was prepared by heating 10.5% fatty-acid free BSA in a water in a 55 °C water bath for 30 min, heating 100 mM palmitate in 100 mM NaOH in a 70 °C water bath until dissolved, and then adding 100 mM palmitate to 10.5% BSA at a 1:19 ratio. The solution was stirred at 37 °C for 30 min and allowed to cool down before being filter sterilized and aliquoted. Aliquotes were kept at −20 °C and warmed to 37 °C before use. Control cells were cultured at 10 mM glucose. Mannitol (at 10 mM) was used as an osmotic control for the incubations at 20 mM glucose. After 48 h, RNA was isolated and used for quantification of mRNA levels.

For experiments using freshly isolated rat islets, these were incubated in RPMI medium containing 20 mM glucose or 1 mM palmitate. The medium was replaced by fresh medium after 24 h. RNA was extracted 48 h after incubation start and used for mRNA expression analysis.

**Incubation of INS-1 832/13 cells with valproic acid.** A 250 mM stock of VPA (Sigma-Aldrich) was prepared fresh at the time of use in distilled water and filter sterilized. VPA at 0.1, 0.3 or 1 mM was added to cells at the time of transfection with *Sox5* or control siRNA. Medium was changed to fresh VPA-containing medium the day after transfection, and insulin secretion or RNA isolation was performed 48 h post-transfection as described in the corresponding sections. For electrophysiological measurements fresh medium with or without 0.3 mM VPA was added to cells the day after seeding. The cells were split 24 h later and new medium with or without VPA was added. The cells were used for experiments after a total incubation time of 48 h with or without 0.3 mM VPA.

**RNA interference in INS-1 832/13 cells.** The day before transfection 350,000–400,000 INS-1 832/13 cells were seeded in each well in a 24-well plate. Cells were transfected with RNA interference oligonucleotides using DharmaFECT 1 (Thermo Scientific) according to the manufacturer's description. Silencer Select siRNAs from Life Technologies were used for all genes. For *Sox5*, two additional siRNAs from Sigma-Aldrich and two siRNAs from Thermo Scientific were used. Final oligonucleotide concentration was 30 nM. We used negative control siRNA from the same manufacturer as the active siRNA: Silencer Select Negative Control no. 2 (Life Technologies), Mission siRNA universal negative control # 1 (Sigma-Aldrich) and On-Target plus non-targeting siRNA # 1 (Thermo Scientific), respectively. Assays were performed 48 h after transfection. For immunohis-tochemistry cells were re-seeded after 24 h onto a 0.175 mm thick glass and incubated for another 24 h. Transfection efficiency was assessed by qPCR as described below. For capacitance measurements, cells were re-seeded in new dishes after 48 h and used for experiments at 48–72 h post-transfection.

**RNA interference in EndoC-BH1 cells.** Sox5 was knocked down in EndoC-BH1 cells using BLOCK-iT HiPerform Lentiviral Pol II miR RNAi Expression System (Thermo Fisher Scientific). A pre-miRNA sequence targeting base 128–148 of human Sox5 transcript variant 1 was designed using BLOCK-It RNAi Designer (Thermo Fisher Scientific). This sequence was used to create a pre-miRNA expression cassette, which was cloned into the pLenti6.4/R4R2/V5-DEST MultiSite Gateway vector (Thermo Fisher Scientific) together with a 409 bp sequence of the RIPII promotor (bp −696 to + 12). The pre-miRNA was expressed co-cistronically with EmGFP in the expression cassette to allow detection of RNAi by fluorescence. For the control plasmid, instead of the Sox5 128–148 pre-miRNA sequence, a pre-miRNA hairpin sequence predicted not to target any known vertebrate gene was used (provided with the kit). Lentiviruses were produced at the Vector Unit at Lund University, and viral titres were determined by transducing INS-1 832/13 cells and measuring GFP-positive cells by FACS 72 h post transduction. EndoC-BH1 cells were transduced with Sox5 RNAi lentivirus at 5 MOI at the time of seeding. Medium was changed after 24 h and secretion was performed after 144 h (at this time full knockdown effect was observed).

**Preparation of dispersed human islets for electrophysiological measurements.** Human islets were hand-picked, dispersed into single cells and seeded on plates pre-coated with poly-L-lysine. Cells were transfected with *SOX5* siRNA (targeting bp 1,428–1,448 of human SOX5 transcript variant 1) or negative control siRNA (Lifetech) using DharmaFECT 1 transfection reagent (Thermo Scientific). The BLOCK-It Alexa Fluor Red Fluorescent Oligo was added to allow for visualization of transfected cells. As the Alexa Fluor Red-coupled oligonuclotides are more bulky than the *SOX5* siRNA oligonucleotides, this approach is unlikely to overestimate the transfection rate. Cells were used for experiments 36 h after transfection.

**Electrophysiological measurements.** The electrophysiological measurements were conducted using an EPC-10 patch clamp amplifier with the PULSE software (HEKA, Germany). The plastic Petri dishes were used as the experimental chamber with a plastic insert to reduce the volume to $\sim 0.5$ ml. The dish was continuously perfused at a rate of $\sim 2$ ml min$^{-1}$ at 31–33 °C. Patch pipettes were pulled from borosilicate glass, coated with Sylgard and fire-polished to an average resistance of 4–6 MΩ when filled with pipette solution. The zero-current potential of the pipette was adjusted with the pipette in the bath. Exocytosis was elicited by a train of ten depolarizations from $-70$ to $0$ mV, which was applied to simulate glucose-induced electrical activity. Exocytosis was monitored as increases in cell capacitance using the sine + DC mode of the lock-in amplifier included in the PULSE software and the standard whole-cell configuration. Human β-cells were identified based on their size ($\sim 10$ pF) and/or immunostaining[48]. The extracellular solution consisted of (mM) 118 NaCl, 20 TEACl, 5.6 KCl, 2.6 CaCl$_2$, 1.2 MgCl$_2$, 5.0 HEPES and 5.0 glucose (pH 7.4 with NaOH). The pipette solution in Fig. 4e contained (mM) 125 K-glutamate, 10 KCl, 10 NaCl, 1 MgCl$_2$, 5 HEPES, 3 ATP, 0.1 cAMP, 10 EGTA and 9 CaCl$_2$ (pH 7.2 with KOH). The pipette solution used in the remaining experiments was composed of (mM) 125 Cs-glutamate, 10 CsCl, 10 NaCl, 1 MgCl$_2$, 5 HEPES, 3 ATP, and 0.1 cAMP and 0.05 EGTA, (pH 7.2 with CsOH). All reagents were of analytical grade. Isradipine was added to the extracellular solution at 2 μM. Tetrodotoxin (0.1 μg ml$^{-1}$) was present to block the Na$^+$ currents. The current–voltage relationship was measured by 100 ms depolarizations from the holding potential at $-70$ mV. The depolarization potential was increased stepwise from $-40$ to $+40$ mV.

**Ca$^{2+}$ imaging in INS-1 832/13.** A low-affinity Ca$^{2+}$ indicator, Fluo-5F (Kd = 2.3 μM) (Invitrogen), was used for measuring intracellular Ca$^{2+}$. Cells were loaded with 1 mM Fluo-5 F for 30 min at 37 °C. Cells were first perfused in a Krebs-Ringer bicarbonate (KRB) buffer containing 116 mM NaCl, 4.7 mM KCl, 2.6 mM CaCl$_2$, 1.2 mM KH$_2$PO$_4$, 1.2 mM MgSO$_4$, 20 mM NaHCO$_3$, 16 mM HEPES and 2 mg ml$^{-1}$ BSA, supplemented with 2.8 mM glucose. The cells were stimulated with a 20 mM glucose KRB buffer at room temperature.

Images were acquired by confocal microscopy (Carl Zeiss, Germany) using a × 63 oil immersion objective (NA = 1.25). An argon laser (488 nm) was used to excite the cells and the emitted light was collected using a 500–530 nm bandpass filter. The ratio of fluorescence intensity at each time point ($F_i$) versus the average fluorescence intensity ($F_0$) under pre-stimulatory conditions was determined[49]. The integrated fluorescence signal (area under the curve) was calculated using the function $\sum_{i=0}^{i} 0.5 \times \Delta t \times (R_i + R_{i+1})$, where $\Delta t$ denotes the time interval and $R_i$ the ratio ($F_i/F_0$) at time point $i$.

**Immunostaining INS-1 322/13 cells.** INS-1 832/13 cells were washed twice in PBS and fixed with 3% PFA in PBS for 30 min at 37 °C followed by washing 3 × 10 min with PBS before permeabilization for 30 min with BD Phosphflow Perm buffer III (BD Biosciences). Samples were blocked using 5% normal donkey serum in PBS for 30 min at room temperature. Cells were incubated overnight at 4 °C with antibodies against Ca$_V$1.2 (1:200, code C1603, Sigma-Aldrich), Ca$_V$1.3 (1:100, code ACC 005, Alomone Labs), MafA (1:100, code ab26405, Abcam), MafB (1:100, code IHC00351, Bethyl Labs), Pax6 (1:100, code MAB2237, Millipore) or Nkx6.1 (1:100, code AF5857, R&D Systems) diluted in the blocking solution. All antibodies were from rabbit except Nkx6.1, which was generated in mouse. Immunoreactivity against the rabbit antibodies was detected by a Delight 488-conjugated antibody (1:400, code 715-545-150, Jackson ImmunoResearch). A Cy5-conjugated antibody (1:400, code 715-175-150, Jackson ImmunoResearch) was used to detect the Nk × 6.1 antibody. Cell nuclei were counterstained with Hoechst 34580 (1:500, Life Technologies). Immunofluorescence images were acquired by confocal microscopy and Zen software (Carl Zeiss).

**Immunostaining human pancreatic sections.** Formalin-fixed paraffin-embedded human T2D and control pancreatic sections were deparaffinized and processed for heat-induced antigen retrieval using Retrievit 2 buffer (Biogenex). For immunohistochemistry the following primary antibodies were used: rabbit anti-Sox5 (1:250, code sc-20091, Santa Cruz), guinea pig anti-insulin (1:1,000, DAKO), mouse anti-glucagon (1:2,000, Sigma-Aldrich). Secondary antibodies applied were Cy3-, Cy2- and Cy5-conjugated anti-rabbit, anti-guinea pig and anti-mouse (1:500, Jackson ImmunoResearch), respectively. DAPI dye was used to perform nuclear counterstaining (1:6,000, Invitrogen). Immunofluorescence images were acquired at × 40 magnification using confocal microscopy and Zen software (LSM 780, Carl Zeiss).

**Quantitative PCR.** Islets were homogenized in Qiazol reagent (Qiagen) followed by vortexing. RNA was extracted with chloroform precipitation using the mRNeasy kit (Qiagen). For total RNA extraction from INS-1 832/13 cells, RNeasy Plus Mini kit (Qiagen) was used.

Reverse transcription was performed using either SuperScript III Reverse transcriptase (Life Technologies) or SuperScript VILO cDNA synthesis kit (Life Technologies). Quantitative PCR was performed on a ViiA 7 Real-Time PCR System (Life Technologies) using TaqMan Gene Expression Assays and TaqMan Universal PCR Master Mix (Life Technologies). Relative gene expression was

measured from triplicate average Cq-values normalized to the geometric mean of reference genes. For human islets, *HPRT1* and *B2M* were found to be stable and were used as reference genes. For INS-1 832/13 cells with Sox5 knockdown, *B2m* and *Polr2a* were found to be stable and were used as reference genes.

For experiments with low mRNA abundance, preamplification was performed for 10 or 14 cycles before quantitative PCR using TaqMan PreAmp Master Mix (Life Technologies) and TaqMan Gene Expression Assays at a dilution of 0.05.

In some experiments where data were available from both non-treated and negative control samples, the relative gene expression in treated samples was normalized to the relative gene expression in the non-treated sample. Average value and s.e.m. from all experiments were then normalized to the average of the negative controls. Statistical comparisons were made using paired Student's $t$-test on log2-transformed relative gene expression values.

**Sox5 overexpression.** INS-1 832/13 cells were transiently co-transfected with a custom-made plasmid expressing rat *Sox5* under the CMV promoter (BlueHeron) and a plasmid encoding GFP (BlueHeron) using the Lipofectamine LTX Plus reagent (Life Technologies) according to the manufacturer's manual. Total plasmid concentration was 0.625 μg ml$^{-1}$ and the concentration of Lipofectamine LTX was 4 μl μg$^{-1}$ plasmid.

The ratio of *Sox5* plasmid to GFP plasmid was 1:1 in the capacitance measurements in which GFP was used to identify transfected cells. The capacitance measurements were performed 48–72 h after transfection. For microarray analysis the *Sox5* plasmid was co-transfected with an empty vector instead of the GFP plasmid at a ratio of 1:1,000. RNA for microarray analysis was extracted 48 h post-transfection. For Western blots cells were co-transfected with *Sox5* and GFP plasmid at a ratio of 1:1 or 1:1,000. These samples were lysed 72 h post-transfection. For these experiments, we used cells transfected with the equivalent amount of GFP plasmid alone as controls.

Human islets were transduced with *SOX5* lentivirus or control virus 72 h before insulin secretion measurements or RNA isolation. The sequence encoding human *SOX5* transcript variant 1 (Origene, RC224228) was cloned into a bicistronic lentiviral vector (Sanbio CD630A-1) where the UBC promotor had been replaced by a RIP promotor. Lentiviruses were produced at the Vector Unit at Lund University. Viral titre was determined with Lenti-X qRT-PCR Titration Kit (Clontech). Islets were transduced in a small volume of medium (400–900 μl for 250–600 islets) and after 24 h transferred to new medium. Measurements were performed 72 h after transduction.

Rat islets were transduced with a LVX-TRE3G-*Sox5*-mCherry plasmid. This plasmid was intended to be used as part of the Lenti-X Tet-On 3G Inducible Expression System (Clontech), but in our case the response plasmid induced expression of *Sox5* without the Tet-On 3G transactivator protein. The experimental procedure for transduction of rat islets was the same as for human islets.

**Western blot.** INS-1 832/13 cells were transfected with siRNA or *Sox5* expression plasmid as previously described. Cells were collected 48 h after transfection for siRNA and 72 h after transfection for *Sox5* overexpression. Briefly, cells were washed once with ice-cold PBS, and then lyzed with RIPA buffer (50 mM Tris HCl, 150 mM NaCl, 1% NP-40/Triton X, 0.1% SDS, 0.5% sodium deoxycholate, 2 mM EDTA, 50 mM NaF) supplemented with protease inhibitor (Complete, Roche). Samples were kept on ice and lightly shaken until cells were detached. The lysate was collected and centrifuged at 10,000g for 10 min. The supernatant was either stored at $-80$ °C or used directly for analysis. Approximately 20–30 μg of protein was loaded per well and separated on 10% or 4–15% Mini-PROTEAN TGX precast polyacrylamide gels (Bio-Rad). Proteins were transferred to a PVDF membrane (GE Healthcare) using regular transfer buffer (58 mM Tris, 186 mM glycine, 0.1% SDS, 20% Ethanol). Blocking of membrane and incubation with antibodies was performed in TBST with 5% milk. Primary antibodies used were against MafA (1:200, code ab26405, Abcam), Ca$_V$1.2 (1:400, code C1603, Sigma-Aldrich), Ca$_V$1.3 (1:400, code #ACC-311, Alomone Labs), Pdx1 (1:1,000, code #5679, Cell Signaling Technology), β-actin (1:2,000, clone AC-15, Sigma-Aldrich) and cyclophilin B (1:2,000, code ab16045, Abcam). Incubation with primary antibodies was performed overnight at 4 °C. Secondary HRP-linked antibodies used were anti-rabbit (1:4,000, code #7074, Cell Signaling Technology) and anti-mouse (1:2,000, code P 0447 Dako). Incubation with secondary antibodies was performed overnight at 4 °C or for 2 h at room temperature. SuperSignal West Femto Maximum Sensitivity Substrate (ThermoPierce) was used for visualization of proteins with a CCD camera (AlphaImager from Alpha Innotech or ChemiDoc XRS + from Bio-Rad). The intensity/volume of each protein band was normalized to the intensity/volume of reference protein (β-actin or cyclophilin B) for each sample. Paired Student's $t$-test was used for statistical analysis. The original pictures of the blots displayed in this paper are shown in Supplementary Fig. 9.

**cAMP measurements.** Intracellular levels of cyclic AMP (cAMP) were measured with the cAMP Biotrak Enzyme Immunoassay System (GE Healthcare Life Sciences) using the non-acetylation protocol as described by the manufacturer. INS-1 832/13 cells (300,000 cells per well) were seeded in a 24-well plate, transfected with *Sox5* siRNA the following day and lysed with 1.5 ml Lysis reagent 1B 48 h post-transfection.

**Proinsulin-insulin ratio.** The ratio of human proinsulin to human insulin in the secretion buffer of INS-1 832/13 cells was determined using Mercodia Proinsulin ELISA and Mercodia ELISA (Mercodia) according to the manufacturer's instructions. Human insulin and proinsulin were analysed since INS-1 832/13 cells, in contrast to other INS-1 clones, are stably expressing the human proinsulin gene. Insulin secretion in Sox5-kd cells was performed 48 h post-transfection, and an aliquot of the secretion buffer was used for proinsulin analysis.

**INS-1 832/13 cell viability.** Cell viability was measured using a CellTiter 96 Aqueous One Solution Cell Proliferation Assay Reagent (Promega) according to the manufacturer's instructions. The measurement is based on the spectrophotometric detection of a coloured formazan product converted from a (3-(4,5-dimethylthiazol-2-yl)-5-(3-carboxymethoxyphenyl)-2-(4-sulfophenyl)-2H-tetrazolium) (MTS) compound by NADPH or NADH in metabolically active cells. Cells were seeded at 50,000 cells per well in a 96-well plate the day before transfection. Six hours after transfection, the medium was changed to a medium containing 0.5 mM palmitate and/or 20 mM glucose. After 48 h incubation, the cells were incubated for 1 h 30 min with CellTiter 96 Aqueous One Solution reagent before measuring the absorbance at 490 nm with a 96-well plate reader.

**Metabolomics.** Metabolite profiling in INS-1 832/13 was performed as previously described in detail[50,51]. In brief, cells were treated as described for insulin secretion, followed by a quick wash in ice-cold PBS and quenching of metabolism by addition of 70 μl ice-cold double distilled water. Metabolites were extracted using a one phase extraction protocol, as previously described in detail[50]. Metabolites were derivatized and analysed on a gas chromatograph (GC; Agilent 6,890 N, Agilent Technologies) connected to a time-of-flight mass spectrometer (TOFMS; Leco Pegasus III TOFMS, Leco Corp., USA). Data were acquired using Leco ChromaTof (Leco Corp.), exported as NetCDF files, and processed using hierarchal multivariate curve resolution[52] in MATLAB 7.0 (Mathworks, Natick, USA). Metabolites were normalized using the scores from the first component of a principal component analysis performed in Simca P+ 12.0 (Umetrics, Sweden) on the uncentreed and unit variance scaled areas of internal standards[53].

**Mitochondria spare respiratory capacity measurements.** OCR were measured by the XF24 Extracellular Flux Analyzer (Seahorse Bioscience), as described by Malmgren et al.[54] and Brand et al.[55]. INS-1 832/13 cells were seeded onto poly-D-Lysine coated XF24 24-well plates at 70,000 cells per well, transfected with Sox5 siRNA and incubated for 48 h before the assay. The cells were preincubated in 500 μl assay buffer (114 mM NaCl, 4.7 mM KCl, 1.16 mM MgSO₄, 2.5 mM CaCl₂, 0.2% bovine serum albumin, 2.8 mM glucose, pH 7.2) for 2 h at 37 °C in air after which basal respiration was measured in the presence of 2.8 mM glucose. After the assessment of basal respiration, 16.7 mM glucose or 10 mM pyruvate was added and OCR was analysed during ∼1 h. Oligomycin (4 μg ml$^{-1}$), the ionophore carbonyl cyanide-p-trifluoromethoxy-phenylhydrazone (FCCP) (4 μM) and rotenone (1 μM) were added in sequence to inhibit ATP synthase, uncouple the inner mitochondrial membrane proton gradient dissipation from ATP synthesis, and to block the electron transport chain, respectively. Data were normalized to basal respiration. Student's t-test was used for statistical analysis.

**Isolation of β cell RNA for gene expression analysis.** β cells from Wistar rat islets were isolated, fixed, labelled intracellularly with insulin antibody and sorted using the method describes by Hrvatin et al.[56]. Briefly, islets were dispersed by pipetting up and down after incubation with TrypLE Express (ThermoFisher Scientific) at 37 °C. The cells were passed through a strainer (50 μm), washed with PBS, and fixed for 30 min with DEPC-treated PBS containing 4% PFA, 0.1% saponin (Sigma-Aldrich) and 1:20 RNasin Plus Rnase inhibitor (Promega). This and remaining steps were performed at 4 °C using RNase-free equipment. The fixed cells were centrifuged at 3,000g for 3 min, washed with DEPC-treated PBS containing 0.2% BSA, 0.1% saponin and 1:100 RNasin Plus, strained through a 50 μm filter, and washed again. The cells were then incubated rotating for 30 min with guinea pig anti-insulin primary antibody (1:1,000, code #2263B65-1, Europroxima) in DEPC-treated PBS containing 1% BSA, 0.1% saponin and 1:20 RNasin Plus. After 2× wash, the cells were incubated rotating for 30 min with FITC-conjugated donkey anti-guinea pig IgG secondary antibody (1:100, code 706-095-148, Jackson ImmunoResearch). The cells were washed twice and suspended in DEPC-treated PBS containing 0.5% BSA and 1:20 RNasin Plus for FACS analysis. The cells were sorted into a β-cell fraction and a β-cell-depleted fraction on a BD FACS Aria II cell sorter. RNA was purified using the RecoverAll Total Nucleic Acid Isolation Kit (Ambion), and SuperScript VILO cDNA synthesis kit was used for reverse transcription of isolated RNA.

**Statistical analyses.** Linear regression was used for comparisons between gene expression in the microarray and continuous phenotypes and logistic regression was used to analyse the association between gene expression and T2D status. The regression analyses were corrected for age, sex and BMI. Fisher's Exact test with Benjamini–Hochberg multiple testing correction was used for enrichment analyses.

A linear model was used for the association analyses with the risk score and the continuous traits.

Student's t-test was used when comparing two experimental groups unless otherwise specified. Data from human islets were analysed using linear regression using average insulin secretion from each individual. The single-cell data, including electrophysiological recordings and granule distribution analysis, were analysed using a linear model in which all single-cell recordings were included as discrete observations but were grouped using donor ID as the subject cluster variable[5]. β values for associations were obtained from the linear model.

Complete phenotype data on insulin secretion, electrophysiology and granule distribution were not available from all human donors included in the study, which is why the number of donors used for the different trait analyses vary (see 'n' for the different series).

Two-sided tests were used unless otherwise specified. One-sided tests were used for the replication analyses and analyses of one-sided hypotheses as specified in the text. All statistical comparisons for animal- and cell experiments were performed using Student's t-test.

Details on the statistical procedure used to analyse data from specific methods are given in the relevant sections. Statistical analyses were performed using R, python 2.6 (additional packages: scipy 0.7.0, fisher 0.1.0 and statsmodels 0.4.0), IBM SPSS Statistics (ver 20.0, 21.0 or 22.0) or Excel.

**Data availability.** All human islet microarray data are MIAME compliant, and the raw data have been deposited in a MIAME database (GEO, accession number: GSE38642).

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

## Acknowledgements

Supported by the Ragnar Söderberg Foundation, the Wallenberg Centre for Molecular and Translational Medicine in Gothenburg, the Swedish Foundation for Strategic Research, the Hjelt Foundation, the Royal Physiographic society, the Albert Påhlsson foundation, the NovoNordisk Foundation, the Swedish Research Council through research positions (L.E.), a Linnaeus grant and a Strategic Research Grant (Exodiab), and the Interdisciplinary Center for Clinical Research (IZKF) Faculty of Medicine at the RWTH Aachen. We thank Dr Raphael Scharfmann for granting us the right to use and providing us with EndoC-BH1 cells, and Britt-Marie Nilsson and Anna-Maria Veljanovska Ramsay for expert technical assistance.

## Author contributions

A.S.A., H.A.N. and T.M. performed most of the experiments, co-designed the study, analysed data and co-wrote the manuscript. B.M., S.H., I.G.C., J.M.J.D., J.M., X.Y., B.Z., E.G.G. and A.H.R. performed bioinformatics analyses. T.S., A.W., A.B., T.M.R., L.S., M.S., Y.T., J.W., S.S., M.F., L.E., I.A., J.D.J., E.Z. and P.S. assisted with the cellular and expression experiments. C.B.W. and H.M. discussed the findings and co-wrote the manuscript. A.H.R. conceived the study, performed bioinformatics analyses and wrote the manuscript.

## Additional information

**Competing interests:** The authors declare no competing financial interests.

