## [Peer review file · Nature Communications]

Reviewers' comments:

Reviewer #1 (Remarks to the Author):

Sox5 regulates beta-cell state and is reduced in type 2 diabetes

The authors analyzed global gene expression in human islets and identified a group of co-expressed genes ('module') that was enriched for genes with islet-selective open chromatin. The authors found one eigengene representing a module of 3032 genes that correlated with T2D, HbA1c and insulin secretion. They called this module the T2D associated module. Genes enriched in this module included ones associated with the pancreas, vesicle release and secretory function. The authors hypothesized that this module contributes to maintenance of β -cell function. They also found that the T2D module contained 168 genes located to regions with islet-selective open chromatin. These genes included critical transcription factors, synaptic function, ion channel activity and insulin secretion. The authors also identified Sox5 as a regulator of the module and Sox5 knockdown induced gene expression changes similar to those observed in T2D and diabetic animals. Sox5 knockdown also had effects on insulin secretion, including reduced depolarization-evoked Ca^{2+} -influx and β -cell exocytosis. The authors then go on to show that Sox5 overexpression or the histone deacetylase inhibitor valproic acid can both improve insulin secretion in T2D human islets. The authors suggest that human T2D islets display changes reminiscent of dedifferentiation and highlight valproic acid as a potential antidiabetic treatment that improves β -cell function through increased expression of Sox5. The authors have done a good job integrating multiple data sets to support their conclusions. Overall this is nice paper, however, there are concerns with some of the interpretation of the results and there are some additional studies required as discussed below.

Major comments:

1. Since Sox genes are often thought to be responsible for male differentiation on the Y chromosome did the authors find any sex differences in their data.
2. The authors suggest that their T2D module is associated with immature β -cells and loss of secretory function. They determine this by looking for overlap with the T2D signature genes with two different data sets, one looking at artificially dedifferentiated human islets and another study looking at pdx1 and insulin gene expression. These two data sets were used to support their conclusion that T2D may be partly due to β -cell dedifferentiation. Although this is interesting the authors overstate the strength of this conclusion. The proposed dedifferentiated state needs to be confirmed in human T2D islets.
3. The rationale for why Sox5 was picked over the other potential key regulators of the T2D-associated module was hard to follow. In Supplemental fig 1a the authors looked at potential key regulators of this module and suggested that Sox5, Lpar1, Ptch1, Tmem196, and Tmem63C mRNA were reduced by about 50%. No stats were provided nor was there a discussion about some of the differences seen between high glucose vs. palmitate. Also, in supplemental figure 1b, only two of five siRNA against Sox5 inhibited insulin secretion (not 3 as suggested in the paper) and one increased insulin secretion. The authors assessed the role of the other potential regulators using two different siRNAs and said they did not affect insulin secretion. However, considering three of the siRNA against Sox5 had no effect on

insulin secretion (or increased insulin secretion) this could have also happened with the two siRNA tested against some of these other key regulators. Since the connection between the authors T2D associated module and its proposed key regulator Sox5 is central to this paper further clarification is required here.

4. The insulin secretion and metabolic studies seen in figs 2-4 were primarily done in INS1 832/13 cells using the most effective inhibitory siRNA against Sox5 from supplemental fig 1 b. Based on the points made above this may not be the best siRNA to use for these studies. Also if Sox5 is involved in the dedifferentiation of islets than using immortalized INS1 cells for these studies is not ideal.

5. The reduced Sox5 in T2D human islets shown in supplemental figure 2d was small. Protein levels need to be confirmed for Sox5.

Minor comments:

1. There are some minor typos that should be fixed throughout the paper.

Reviewer #2 (Remarks to the Author):

In the present study, Rosengren and co-workers analyzed global gene expression in human islets from control and type 2 diabetic patients (T2D) and found a group of co-expressed genes (module) regulated by Sox5. Functional experiments based on KD or overexpression in different experimental models confirmed a role for Sox5 in insulin secretion and associated processes, and indicated that the anti-epileptic agent valproic acid improves beta cell function via increased expression of Sox5.

This is an interesting study, showing the power of combining sophisticated analysis of functional genomics data from human islets obtained from control or T2D individuals with detailed functional studies. Of concern, however, several methodological questions must be solved to support the main conclusions of the study. Some of these issues (described below) will require additional experiments.

Specific comments

1. I suggest changing the title from "Sox5 regulates beta cell state and is..." for "Sox5 regulates beta cell phenotype and is...".
2. It is mentioned in Methods that average values for insulin release are based on "technical replicates" (2-8 wells) from 3 or more experiments (actually, in some cases of only one experiment; see below). Technical replicates from the same experiment, i.e. cell passage or islet isolation should be considered as $n=1$, and a mean made based on the independent experiments performed, i.e. 3 independent experiments with 7 "technical replicates" each is $n=3$, and not $n=21$.
3. Relative gene expression was normalized by the geometric means of Hprt1, B2m and Polr2a cq values. Did the authors exclude that expression of these housekeeping genes is affected by the different experimental conditions used? This should be indicated in the text.
4. Information should be provided on the MOI used for transduction of human islets with the Sox5 lentivirus. Were the islets dispersed ahead of viral transduction?
5. Please indicate the efficiency of transfection of INS-1 cells with the Sox5 expression plasmid.
6. It is mentioned in page 33 (Suppl. Experimental Procedures) that "...the ratio of human proinsulin to human insulin in ...INS-1...was determined...". Is this correct, or should it be rat proinsulin to insulin rate? If correct, why use a human kit to measure a rat hormone?
7. Have the same human islet preparations used in the present study, from both control and T2D individuals, also used in previous studies from the same group? If yes, this should be indicated, and information provided on the novelty of the present analysis as compared to previous ones.
8. Figures 1c-e are apparently not mentioned in the text of Results. Please double check.
9. Please indicate the number of independent experiments performed in Suppl. Fig. 1a (i.e. islets isolated from x rats), Figure 3C, Suppl. Figure 3A and Figure 5.
10. In Suppl. Fig1 c, it should be written Sox5-kd and not Sok5-kd.
11. The information provided in Fig. 1b is not clear. There appears to be a decrease in insulin secretion by siSox5 1 and 5, and an increase in insulin release by siSox 3. This is not in line the description of the Figure provided in Results.
12. It is mentioned in Results that the "overall viability was similar between Sox5kd and control cells" (Suppl. Fig. 1). This is based on a single experiment ($n=1$), with 5-6 technical replicates, performed under basal (non-stressed) conditions. It is crucial to re-evaluate

viability of Sox5kd cells after 24-48 exposure to palmitate or high glucose in at least 3 independent experiments. It is indeed possible that Sox5kd will only affect viability under metabolic stress, a finding of direct relevance for the T2D situation.

13. In Suppl. Figures 3C, 3H and 5C data are "technical replications" from a single experiment (n=1). Please confirm these observations by additional experiments.

14. In Suppl. Figs 2e and 2f, the data shown indicates correlation between docked granules and Sox5 mRNA expression in beta and alpha cells from T2D patients. Is the same correlation observed in non-diabetic individuals?

15. In Suppl. Figure 3g, Sox5 is overexpressed by use of a lentiviral vector, but the legend of the figure mentions "Sox5-KD". Please correct.

16. Figure 4h show crucial findings in human beta cells. The information must be complemented by glucose-stimulated insulin release (n=3, at least).

17. Culture of human islets at 20 mM for 3 days did not affect module genes (Fig. 5f), but it would be important to compare these findings against available data on the impact of the saturated FFA palmitate on mRNA expression in human islets (Cnop M et al., 2014, PMID: 24379348).

18. The blood glucose of the mice treated or not with phlorizin (Fig. 5) must be provided.

19. There is one figure missing in Figure 6 - the description mentions a figure 6.i., which is not present. Please double check the Figure, Figure legend and description of the Figure in Results.

20. It would be important to provide an additional Supplementary Table indicating number of experiments and the nature of the 81 of the 168 genes changed by VPA. This would allow evaluation on whether these genes are among the important ones for the maintenance of beta cell phenotype.

21. The Authors have access to 3 islet preparations from T2D patients, as shown in Figure 7b. It is thus difficult to understand why they studied the effects of VPA - a key experiment for the paper - in a single T2D patient in Figure 7d. These findings should be reproduced in additional T2D islets.

Reviewer #3 (Remarks to the Author):

Summary: Axelsson, AS, et al. have utilized a bioinformatics approach to elucidate gene co-expression networks from type 2 diabetic donors in comparison to normal donors. Genes grouped in co-expressed modules were in open chromatin regions (identified by FAIRE-Seq in previous study), that were found to be differentially expressed in T2D islets. One gene identified in this diabetic module, Sox5, was further characterized in rat insulinoma cell lines (Ins-1 832/13). Gene expression changes in db/db and ob/ob islets were correlated to changes in Sox5 levels.

Overall comments: This manuscript is possibly clearly written for readers whose background is strong in bioinformatics. However, the language at the beginning of the manuscript is very difficult to comprehend and not well explained to the non-bioinformaticists. The reviewer needed to look up papers to understand what the authors were trying to get across. In addition, the manuscript changes abruptly from experiment to experiment with little introduction or flow. The identification of Sox5 as a novel regulator of islet beta-cell function is intriguing, but the majority of work was performed in an immortalized rat beta cell line. Care should be taken in interpretation of these results without confirmation in other systems. Several specific concerns outlined below need to be addressed.

Specific comments:

1. Why was the microarray data from the "replication set" of 59 donor islets not used as the primary analysis but was instead examined after the initial investigation of 64 global microarrays from human donors? Data from islets would be more pertinent to the studies of beta cell function in the rest of the manuscript.
2. Identification of "key regulators" of the T2D-associated module were based upon the response of rat islets to 48hrs of high glucose or palmitate. This is a short-term treatment compared with long-term disease of T2D and was tested in rat versus human islets. This should be considered when attributing gene dysfunction to decreases in Sox5 expression especially considering the very subtle decrease in SOX5 expression in T2D (Sup figure 2d).
3. What variations in Sox5 expression are there across individual beta cells from T2D donors and between different T2D donors? Are there some beta cells with high and some with low Sox5 expression? Moreover, what are the levels of Sox5 in islet alpha and delta cells from normal and T2D donors? IHC with Sox5 specific antibody, or FISH with Sox5 probe could answer these questions.
4. The changes in docked granules in T2D donors with low or high Sox5 expression in islet is correlative, as most of these islets will have reduced levels of several islet-enriched transcription factors.
5. How was the transfection of siRNA for SOX5 in human beta cells performed in figure 4h? Through lipid-based methods? How efficient was the knockdown?
6. In Figure 5, the CDF plots are difficult to interpret. Which of the 168 genes that are being monitored have the most significant change between groups? Are all of the 168 genes altered in a predictable way depending on the condition? In addition, the comparison of mouse models to T2D in humans can only be correlated (at best) with changes in Sox5 expression as many other genes/pathways are also affected in these animal models.
7. In figure 5f it is not surprising that culturing human islets in high glucose for 3 days does not correlate with the T2D module, since T2D occurs over decades and involves multiple organ systems.

8. In Figure 6C, if Palmitate reduces Yy1 mRNA levels in a dose dependent manner, and Yy1 positively controls Sox5 levels, shouldn't the treatment of Ins-1 cells with increasing amounts of Palmitate and Yy1-siRNA more significantly reduce Sox5 levels? Please clarify. Additionally there is no discussion concerning the multiple putative regulators that led to a more significant up-regulation of Sox5 mRNA compared with the very minor decrease as a result of Yy1 knockdown.

9. It is not clear why valproic acid was chosen for the experiments in figures 6&7 other than its inhibition of HDACs. Notably, this inhibitor will have a global effects on gene expression and is therefore not specific to Sox5 despite the association between Sox5 expression and VPA dose. How many other genes are also dependent upon VPA dose?

10. The statistics in several figures (3, 6, 7) are hard to believe. Several of the SEM bars are overlapping between control and treatment groups and the p values from the student's t-test are < 0.01 .

Response to reviewers' comments:

Reviewer #1 (Remarks to the Author):

Sox5 regulates beta-cell state and is reduced in type 2 diabetes

The authors analyzed global gene expression in human islets and identified a group of co-expressed genes ('module') that was enriched for genes with islet-selective open chromatin. The authors found one eigengene representing a module of 3032 genes that correlated with T2D, HbA1c and insulin secretion. They called this module the T2D associated module. Genes enriched in this module included ones associated with the pancreas, vesicle release and secretory function. The authors hypothesized that this module contributes to maintenance of β -cell function. They also found that the T2D module contained 168 genes located to regions with islet-selective open chromatin. These genes included critical transcription factors, synaptic function, ion channel activity and insulin secretion. The authors also identified Sox5 as a regulator of the module and Sox5 knockdown induced gene expression changes similar to those observed in T2D and diabetic animals. Sox5 knockdown also had effects on insulin secretion, including reduced depolarization-evoked Ca^{2+} -influx and β -cell exocytosis. The authors then go on to show that Sox5 overexpression or the histone deacetylase inhibitor valproic acid can both improve insulin secretion in T2D human islets. The authors suggests that human T2D islets display changes reminiscent of dedifferentiation and highlight valproic acid as a potential antidiabetic treatment that improves β -cell function through increased expression of Sox5. The authors have done a good job integrating multiple data sets to support their conclusions. Overall this is nice paper, however, there are concerns with some of the interpretation of the results and there are some additional studies required as discussed below.

We thank the reviewer for his/her positive and constructive comments which have been very helpful for us to improve the manuscript.

Major comments:

1. Since Sox genes are often thought to be responsible for male differentiation on the Y chromosome did the authors find any sex differences in there data.

This is an interesting suggestion and have analyzed sex differences across the human data (only male animals were used for rodent studies). However, we do not find any signs of sex influence on the associations. There is no significant difference in SOX5 expression between male and female islet donors. Adjusting the regression analyses for sex differences did not significantly change the associations. This information has now been added to page 12.

2. The authors suggest that their T2D module is associated with immature β -cells and loss of secretory function. They determine this by looking for overlap with the T2D signature genes with two different data sets, one looking at artificially dedifferentiated human islets and another study looking at pdx1 and insulin gene expression. These two data sets were used to support their conclusion that T2D may be

partly due to β -cell dedifferentiation. Although this is interesting the authors overstate the strength of this conclusion. The proposed dedifferentiated state needs to be confirmed in human T2D islets.

We have analyzed 8 key genes that are well established markers of differentiated beta-cells. These genes include the transcription factors PDX1, NKX6.1, MAFA, the glucose transporter GLUT2 (SLC2A2), prohormone convertase 1 (PCSK1) that is essential for insulin processing, the GLP1 receptor, the zinc transporter 8 (SLC30A8) and GAD2 that catalyzes GABA production. These genes were all downregulated in human islets from T2D donors compared with non-diabetic donors, in agreement with a dedifferentiation pattern. These data have been added to Supplementary Table 12. We also observed that 6 of these 8 genes were significantly upregulated after SOX5 overexpression in human T2D islets. These new data have been added to Fig. 7d. There is no general definition of de-differentiated beta-cells, but these data show that genes that are essential for maintaining a differentiated beta-cell state, both in terms of gene expression and secretory function, are downregulated in T2D and can be reversed by SOX5 overexpression. Moreover, the T2D signature that we have identified display an expression pattern similar to that observed in islets from old animals (14 months) and db/db mice, two animal models previously used to signify dedifferentiated beta-cells (Talchai et al., Cell 2012).

3. The rationale for why Sox5 was picked over the other potential key regulators of the T2D-associated module was hard to follow. In Supplemental fig 1a the authors looked at potential key regulators of this module and suggested that Sox5, Lpar1, Ptch1, Tmem196, and Tmem63c mRNA were reduced by about 50%. No stats were provided nor was there a discussion about some of the differences seen between high glucose vs. palmitate.

Also, in supplemental figure 1b, only two of five siRNA against Sox5 inhibited insulin secretion (not 3 as suggested in the paper) and one increased insulin secretion. The authors assessed the role of the other potential regulators using two different siRNAs and said they did not affect insulin secretion. However, considering three of the siRNA against Sox5 had no effect on insulin secretion (or increased insulin secretion) this could have also happened with the two siRNA tested against some of these other key regulators. Since the connection between the authors T2D associated module and its proposed key regulator Sox5 is central to this paper further clarification is required here.

We have now replicated these analyses in two additional series of islets from Wistar rats. These data show that SOX5, PTCH1, SMARCA1, TCF3, TMEM196 and TMEM63C are consistently downregulated both in response to high palmitate and glucose. The figure has been updated with the additional data. It is now also pointed out in the discussion that we cannot exclude that these other genes may also regulate beta-cell function, as there are often many different regulators acting in concert. However, we decided to focus on SOX5 because SOX5 silencing reduced insulin secretion and because SOX5 was indicated as a putative regulator of the module using two independent bioinformatics methods (both transcription factor enrichment analysis and expression-SNP data analysis), as opposed to the other putative regulators which each was indicated by only one method.

The reviewer is correct on his/her comment on Suppl. Fig. 1b, and the text has been changed accordingly. We have no clear explanation for why insulin secretion was not reduced by all different Sox5 siRNAs that were tested, but it might be attributed to uncharacterized effects on splicing due to the siRNAs targeting sequences within or outside the functionally important coil region. The role of Sox5 for beta-cell function is however further corroborated by our demonstration that human beta-cell exocytosis is reduced following transfection of siRNA against SOX5 and that SOX5 overexpression in

human T2D islets improves glucose-stimulated insulin secretion. Moreover, in the revised version we have also added data showing that siRNA-mediated knock-down of SOX5 in human EndoC-BH1 cells reduces insulin secretion (Fig. 7A).

4. The insulin secretion and metabolic studies seen in figs 2-4 were primarily done in INS1 832/13 cells using the most effective inhibitory siRNA against Sox5 from supplemental fig 1 b. Based on the points made above this may not be the best siRNA to use for these studies. Also if Sox5 is involved in the dedifferentiation of islets than using immortalized INS1 cells for these studies is not ideal.

In addition to the siRNA used in INS1 832/13 cells, we have in the revised version also studied the effects of SOX5 knock-down in human EndoC-BH1 cells using human RNAi constructs against SOX5, which corroborate the INS1 832/13 data by producing a 29% reduction of glucose-stimulated insulin secretion (Fig. 7A). We have also added data that viral SOX5 overexpression in human T2D islets increased the levels of key beta-cell genes (Supplemental Table 12). These new experiments together with the previous observations, which demonstrate an inhibitory effect on human beta-cell exocytosis after transfection with a human siRNA against SOX5 and improved insulin secretion in T2D islets after lentivirus-mediated SOX5 overexpression, means that we now have several lines of data using different techniques in both cell lines and human cells which support that SOX5 downregulation or overexpression have profound functional and expression effects.

5. The reduced Sox5 in T2D human islets shown in supplemental figure 2d was small. Protein levels need to be confirmed for Sox5.

In the revised version we have added immunohistochemistry data from pancreatic sections which show that nuclear SOX5 protein was reduced by 67% in T2D beta-cells compared with non-diabetic beta-cells (Supplementary Fig. 2).

Minor comments:

1. There are some minor typos that should be fixed throughout the paper.

We have gone through the manuscript in detail to correct typos.

Reviewer #2 (Remarks to the Author):

In the present study, Rosengren and co-workers analyzed global gene expression in human islets from control and type 2 diabetic patients (T2D) and found a group of co-expressed genes (module) regulated by Sox5. Functional experiments based on KD or overexpression in different experimental models confirmed a role for Sox5 in insulin secretion and associated processes, and indicated that the anti-epileptic agent valproic acid improves beta cell function via increased expression of Sox5.

This is an interesting study, showing the power of combining sophisticated analysis of functional genomics data from human islets obtained from control or T2D individuals with detailed functional studies. Of concern, however, several methodological questions must be solved to support the main conclusions of the study. Some of these issues (described below) will require additional experiments.

We thank the reviewer for his/her encouraging comments to our work and constructive suggestions which have helped us improve the manuscript.

Specific comments

1. I suggest changing the title from "Sox5 regulates beta cell state and is..." for "Sox5 regulates beta cell phenotype and is...".

We think this is a good suggestion and have changed the title accordingly.

2. It is mentioned in Methods that average values for insulin release are based on "technical replicates" (2-8 wells) from 3 or more experiments (actually, in some cases of only one experiment; see below). Technical replicates from the same experiment, i.e. cell passage or islet isolation should be considered as $n=1$, and a mean made based on the independent experiments performed, i.e. 3 independent experiments with 7 "technical replicates" each is $n=3$, and not $n=21$.

This has now been updated throughout with additional experiments. However for electrophysiological single-cell experiments it is customary to present each cell as an individual measurement, since the experimental conditions are less standardized, with critical parameters such as cell capacitance, cell resistance and leak current varying considerably between individual cells. We have therefore adhered to that convention for the electrophysiological experiments.

3. Relative gene expression was normalized by the geometric means of Hprt1, B2m and Polr2a cq values. Did the authors exclude that expression of these housekeeping genes is affected by the different experimental conditions used? This should be indicated in the text.

We thank the reviewer for pointing this out. For human islets with SOX5 overexpression HPRT1 and B2M were found to be stable and were used as reference genes. For INS-1 832/13 cells treated with Sox5 siRNA, B2m and Polr2a were found to be stable and were used as reference genes. The corresponding figures have been updated accordingly throughout the manuscript.

4. Information should be provided on the MOI used for transduction of human islets with the Sox5 lentivirus. Were the islets dispersed ahead of viral transduction?

The islets were undispersed when they were transduced with lentivirus. We titrated the dose to reach optimal degree of overexpression, and used between 1.5 to 8 million viral copies/islet for the different experiments, to reach on average 2.8 ± 0.4 -fold overexpression in the human islets.

5. Please indicate the efficiency of transfection of INS-1 cells with the Sox5 expression plasmid.

We had an 11.6 ± 2.6 -fold overexpression of Sox5.

6. It is mentioned in page 33 (Suppl. Experimental Procedures) that ..."the ratio of human proinsulin to human insulin in ...INS-1...was determined...". Is this correct, or should it be rat proinsulin to insulin rate? If correct, why use a human kit to measure a rat hormone?

We measured the total content of proinsulin relative to insulin content in the cells. The INS1-831/13 cells express the human insulin gene, hence the human kit for the measurements. We have clarified this in the methods section.

7. Have the same human islet preparations used in the present study, from both control and T2D individuals, also used in previous studies from the same group? If yes, this should be indicated, and information provided on the novelty of the present analysis as compared to previous ones.

Some of these human islets microarray data, but not the full cohort used here, have been utilized in other studies from Lund University Diabetes Centre (Mahdi et al., Cell Metabolism 2012, Taneera et al. Cell Metabolism 2012 and Rosengren et al., Diabetes 2012). It is natural that these precious global gene expression data sets are reused in different studies, just as researchers often use patient cohorts for multiple types of analyses (the hundreds of publications on the UKPDS material is one example). The identification of the T2D-associated module and its characterization from both expression and function viewpoints as well as the SOX5 characterizations in human islets are completely novel. We have commented on this in the manuscript on page 28.

8. Figures 1c-e are apparently not mentioned in the text of Results. Please double check.

These figures are referred to on page 7.

9. Please indicate the number of independent experiments performed in Suppl. Fig. 1a (i.e. islets isolated from x rats), Figure 3C, Suppl. Figure 3A and Figure 5.

This has all been updated in the corresponding figure legends.

10. In Suppl. Fig1 c, it should be written Sox5-kd and not Sok5-kd.

This has now been corrected.

11. The information provided in Fig. 1b is not clear. There appears to be a decrease in insulin secretion by siSox5 1 and 5, and an increase in insulin release by siSox 3. This is not in line the description of the Figure provided in Results.

The reviewer is correct and this has now been clarified in the text on page 10.

12. It is mentioned in Results that the "overall viability was similar between Sox5kd and control cells" (Suppl. Fig. 1). This is based on a single experiment (n=1), with 5-6 technical replicates, performed under basal (non-stressed) conditions. It is crucial to re-evaluate viability of Sox5kd cells after 24-48 exposure

to palmitate or high glucose in at least 3 independent experiments. It is indeed possible that Sox5kd will only affect viability under metabolic stress, a finding of direct relevance for the T2D situation.

These data have now been replicated also in response to palmitate and high glucose, which support our previous observation that viability is unaffected.

13. In Suppl. Figures 3C, 3H and 5C data are "technical replications" from a single experiment (n=1). Please confirm these observations by additional experiments.

Suppl. Figure 3C refers to electrophysiology data for which it is customary to report number of cells (see also response to point 2).

Suppl. Figure 3h is now Suppl. Figure 5g, and has been updated with more series (n=3).

Suppl. Figure 5b and 5c have been replaced by a new figure, Suppl. Figure 7b, and have been updated with more series (n=3).

14. In Suppl. Figs 2e and 2f, the data shown indicates correlation between docked granules and Sox5 mRNA expression in beta and alpha cells from T2D patients. Is the same correlation observed in non-diabetic individuals?

As suggested by the reviewer we have now also analyzed cells from 11 non-diabetic donors and found no significant correlation between docked granules and SOX5 mRNA in non-diabetic donors. This observation has been added to the manuscript on page 13.

15. In Suppl. Figure 3g, Sox5 is overexpressed by use of a lentiviral vector, but the legend of the figure mentions "Sox5-KD". Please correct.

This has been corrected, and the figure is now Suppl. Fig. 5h.

16. Figure 4h show crucial findings in human beta cells. The information must be complemented by glucose-stimulated insulin release (n=3, at least).

We have tried very hard to effectively infect human islets with lentiviruses to enable knock-down of SOX5. However, gene knock-down in human islets is well-known to be difficult. During the revision period we tried with preparations from totally 8 donors using varying timing after transfection (72-144 h), 3 different virus constructs and a range of virus titer of 5-40 MOI without observing effective knock-down (on average $15 \pm 5\%$). Using fluorescent Alexa555-coupled oligonucleotides we have observed that only the outermost cell layers take up constructs while the uptake in the islet core is very low (please see the figure below composed of a 3D stack of confocal images which depicts a human islet with Alexa555-coupled oligonucleotides). The low penetration into the islet core means that the functional effect of Sox5 knock-down is likely to be diluted in intact islets. However, when we make single-cell capacitance recordings on human beta-cells with high fluorescence we observe a 50% reduction of beta-cell exocytosis.

In face of this technical hurdle, we used human EndoC-BH1 cells as an alternative option to investigate the effect of SOX5 knock-down on glucose-stimulated insulin secretion in insulin-secreting cells of human origin. SOX5 expression was reduced by 74% in EndoC cells infected by anti-SOX5 lentivirus compared with control virus. This resulted in a 29% reduction of glucose-stimulated insulin secretion ($p < 0.01$). These new data have been added to Figure 7A.

Taken together, several experimental series using different cellular systems and techniques now support that SOX5 expression affects insulin secretion: 1) Sox5 knock-down in INS1-832/13 cells reduces insulin secretion, 2) Sox5 knock-down in human EndoC-BH1 cells impairs insulin secretion, 3) Sox5 knock-down in single human beta-cells reduces insulin exocytosis and 4) SOX5 overexpression in T2D islets increases insulin secretion.

17. Culture of human islets at 20 mM for 3 days did not affect module genes (Fig. 5f), but it would be important to compare these findings against available data on the impact of the saturated FFA palmitate on mRNA expression in human islets (Cnop M et al., 2014, PMID: 24379348).

We thank the reviewer for this suggestions, which led us to analyze these published data to investigate the effect of palmitate (48 h) treatment on the signature. The analysis showed that 16 out of the 169 signature genes were significantly changed as a result of palmitate treatment. Of these, 7 had expression changes in the same direction as in T2D islets and in response to Sox5-knockdown. Nine of them changed in opposite expression to that observed in T2D islets and in response to Sox5-knockdown. Hence, there is no support that the signature would change in response to short-term palmitate treatment (although we cannot exclude that longer treatment would affect a larger fraction of the signature). This has now been added to the manuscript on page 19.

18. The blood glucose of the mice treated or not with phlorizin (Fig. 5) must be provided.

Non-fasted blood glucose was measured before sacrifice and showed

db/db 4 w: Ctrl 4.8 ± 0.6 mM; after phlorizin 4.7 ± 0.4 mM

db/db 10 w: Ctrl 20.0 ± 3.2 mM, after phlorizin 8.4 ± 1.5 mM.

This information has now been added.

19. There is one figure missing in Figure 6 - the description mentions a figure 6.i., which is not present. Please double check the Figure, Figure legend and description of the Figure in Results.

This is a mistyping which has now been corrected. We thank the reviewer for observing this.

20. It would be important to provide an additional Supplementary Table indicating number of experiments and the nature of the 81 of the 168 genes changed by VPA. This would allow evaluation on whether these genes are among the important ones for the maintenance of beta cell phenotype.

A Supplementary Table of these genes (from 3 independent experiments) have now been added (Supplementary Table 11). The 81 genes contained several key genes for beta-cell differentiation and function (e.g. ISL1, NKX6.1 and SLC30A8), and were enriched for membrane proteins (48 of the genes; $p=3E-5$).

21. The Authors have access to 3 islet preparations from T2D patients, as shown in Figure 7b. It is thus difficult to understand why they studied the effects of VPA - a key experiment for the paper - in a single T2D patient in Figure 7d. These findings should be reproduced in additional T2D islets.

Human islet material is scarce, and this paper contains an unusual amount of human islet data, both in terms of expression and functional investigations. However, for some series, especially those using islets from T2D donors we have been forced to prioritize how to use the limited material. For human T2D islets, we prioritized SOX5 overexpression experiments (Fig. 7), as we considered them to be essential for the manuscript. From one T2D donor, however, we had extra material which we used to analyze the effect of VPA on insulin secretion. This analysis was reported in the previous figure 7D.

We have unfortunately not been able to replicate these findings in additional T2D donors during the revision period due to severe shortage of material from T2D donors. We understand the reviewer's request for 3 different T2D preparations and are therefore willing to remove the figure showing VPA effects on insulin secretion in human T2D islets. The abstract, figure legend and manuscript text have been revised accordingly, while all other VPA data from cell lines, mouse islets and mice in vivo remain as in the previous version. We have also replicated the mouse data in Supplementary Figure 7b using 2 additional series (making $n=3$ independent experiments), which further corroborate the previous results showing that VPA increases insulin exocytosis and secretion in INS831/13 cells, insulin secretion in mouse islets and plasma insulin in mice in vivo.

Reviewer #3 (Remarks to the Author):

Summary: Axelsson, AS, et al. have utilized a bioinformatics approach to elucidate gene co-expression networks from type 2 diabetic donors in comparison to normal donors. Genes grouped in co-expressed modules were in open chromatin regions (identified by FAIRE-Seq in previous study), that were found to be differentially expressed in T2D islets. One gene identified in this diabetic module, Sox5, was further characterized in rat insulinoma cell lines (Ins-1 832/13). Gene expression changes in db/db and ob/ob islets were correlated to changes in Sox5 levels.

Overall comments: This manuscript is possibly clearly written for readers whose background is strong in bioinformatics. However, the language at the beginning of the manuscript is very difficult to comprehend and not well explained to the non-bioinformaticists. The reviewer needed to look up papers to understand what the authors were trying to get across. In addition, the manuscript changes abruptly from experiment to experiment with little introduction or flow. The identification of Sox5 as a novel regulator of islet beta-cell function is intriguing, but the majority of work was performed in an immortalized rat beta cell line. Care should be taken in interpretation of these results without confirmation in other systems. Several specific concerns outlined below need to be addressed.

We thank the reviewer for his/her constructive comments which have been very useful for us to improve the manuscript. We have also tried to better introduce the bioinformatics analyses.

Specific comments:

1. Why was the microarray data from the "replication set" of 59 donor islets not used as the primary analysis but was instead examined after the initial investigation of 64 global microarrays from human donors? Data from islets would be more pertinent to the studies of beta cell function in the rest of the manuscript.

The two sets of microarray data were gathered over time. When we did the first analysis we had access to data from 64 donors. These analyses were used to identify the module, the gene signature and SOX5 as a regulator of the module. Over the extended time period when we functionally characterized SOX5 we gathered an additional 59 donors for microarray analysis. These were then used as a replication set, which confirms that the signature is affected in T2D islets also in the replication set. These represent some of the largest cohorts of human islets in the world, and we see the ability to replicate our findings in a second human islet cohort as a major strength of our paper.

2. Identification of "key regulators" of the T2D-associated module were based upon the response of rat islets to 48hrs of high glucose or palmitate. This is a short-term treatment compared with long-term disease of T2D and was tested in rat versus human islets. This should be considered when attributing gene dysfunction to decreases in Sox5 expression especially considering the very subtle decrease in SOX5 expression in T2D (Sup figure 2d).

The identification of potential "key regulators" was done by bioinformatics analysis of human islets (SOX5 was indicated using two independent methods, transcription factor enrichment analysis and human SNP data analysis), but the reviewer is fully correct that the time period of the functional

experiments used to prioritize among the potential regulators is far different from the prolonged T2D state. We have taken the reviewer's comment into account in the revised version and emphasize this on page 10. The lack of good models of T2D is a long-standing problem in the research field, and cultured islets, diabetic animal models, knock-out mice or other artificial simulations of diabetes all have their weaknesses. This manuscript has an unusually strong human relevance in that we start out with a set of human islets from T2D patients to identify the gene module and SOX5 as a putative regulator, and then validate the human relevance of our functional studies by demonstrating an improvement of insulin secretion and increased expression of key beta-cell genes after SOX5 overexpression in human T2D islets.

3. What variations in Sox5 expression are there across individual beta cells from T2D donors and between different T2D donors? Are there some beta cells with high and some with low Sox5 expression? Moreover, what are the levels of Sox5 in islet alpha and delta cells from normal and T2D donors? IHC with Sox5 specific antibody, or FISH with Sox5 probe could answer these questions.

We have analyzed SOX5 expression in pancreatic sections using IHC. SOX5 protein was expressed both in alpha- and beta-cells, and, to a lower degree, in exocrine tissue. We could not observe any expression in delta-cells. Nuclear SOX5 was reduced by 67% in T2D beta-cells compared with non-diabetic beta-cells ($p < 0.001$; Supplementary Fig. 2). There are indeed some beta-cells with high and some with low SOX5 expression, and the standard deviation of SOX5 was considerably higher in normal beta-cells (34 ± 18 a.u.) than in T2D beta-cells (11 ± 5 a.u.). It is therefore of interest that the association between SOX5 expression and insulin secretion was considerably stronger in islets from donors carrying a high number of risk alleles for T2D (see page 26). This suggests that the manifestation of changed SOX5 expression appears to be influenced by DNA variants, such that genetically susceptible individuals develop a more severe secretory failure in response to reduced SOX5 expression.

4. The changes in docked granules in T2D donors with low or high Sox5 expression in islet is correlative, as most of these islets will have reduced levels of several islet-enriched transcription factors.

The reviewer is correct on this notion, and we have included a comment on this on page 13.

5. How was the transfection of siRNA for SOX5 in human beta cells performed in figure 4h? Through lipid-based methods? How efficient was the knockdown?

This was performed by lipofectamine transfection of SOX5-targeting siRNA. Details have been added to the Supplementary Materials and Methods. We could not assess the transfection efficiency since it was based on single cells. Instead, transfection efficiency was assessed by co-transfecting the beta-cells with an Alexa555-fluorescent oligonucleotide, to enable single-cell recordings of cells that were fluorescent, indicating effective transfection. This has now been commented on in the paper on page 15.

6. In Figure 5, the CDF plots are difficult to interpret. Which of the 168 genes that are being monitored have the most significant change between groups? Are all of the 168 genes altered in a predictable way depending on the condition? In addition, the comparison of mouse models to T2D in humans can only be correlated (at best) with changes in Sox5 expression as many other genes/pathways are also affected in these animal models.

We have added two new tables (Supplementary Table 9 and 11) to show which of the signature genes change in response to Sox5-kd, overexpression or VPA treatment. The genes which exhibited consistent expression changes in response to Sox5 perturbation had on average higher connectivity ($p = 0.047$) and

were more strongly correlated with T2D status ($p=1E-6$) compared with genes that were unaffected by Sox5-kd and overexpression. They were also enriched for genes involved in membrane function ($p=0.04$) and ion channel activity ($p=0.03$).

We agree with the reviewer that the comparison of mouse models to T2D in humans can only be correlated with changes in Sox5 expression as many other genes and pathways are also affected, and we now stress this on page 19.

7. In figure 5f it is not surprising that culturing human islets in high glucose for 3 days does not correlate with the T2D module, since T2D occurs over decades and involves multiple organ systems.

This is correct and we have added a comment on this in the manuscript on page 19.

8. In Figure 6C, If Palmitate reduces Yy1 mRNA levels in a dose dependent manner, and Yy1 positively controls Sox5 levels, shouldn't the treatment of Ins-1 cells with increasing amounts of Palmitate and Yy1-siRNA more significantly reduce Sox5 levels? Please clarify. Additionally there is no discussion concerning the multiple putative regulators that led to a more significant up-regulation of Sox5 mRNA compared with the very minor decrease as a result of Yy1 knockdown.

We interpret these observations as a result of an abolished inhibitory effect of palmitate on Sox5 expression after Yy1 knock-down, indicating that palmitate affects Sox5 levels via expression changes of Yy1. On page 20 we have now added a discussion of Nkx2.2 and Sox17, two of the genes for which silencing produced Sox5 up-regulation.

9. It is not clear why valproic acid was chosen for the experiments in figures 6&7 other than its inhibition of HDACs. Notably, this inhibitor will have a global effects on gene expression and is therefore not specific to Sox5 despite the association between Sox5 expression and VPA dose. How many other genes are also dependent upon VPA dose?

VPA was chosen because of its inhibitory action on HDACs and because it has been well studied in other systems in terms of effective doses in vitro and in vivo. However, the reviewer is completely right that VPA has a multitude of effects in addition to Sox5 expression (1348 other module genes were significantly affected), which is now emphasized in the manuscript on page 22. A more detailed investigation of different classes of HDACs on insulin secretion will be of interest for future studies but is beyond the scope of this paper.

10. The statistics in several figures (3, 6, 7) are hard to believe. Several of the SEM bars are overlapping between control and treatment groups and the p values from the student's t-test are < 0.01 .

The reason for this is variation in absolute values between rounds, while the relative effect of the treatment was consistent in each round. The paired t-test takes this into account, and the effect of the treatment is significant even though the SEM error bars are overlapping (due to round-to-round variation of the absolute values).

REVIEWERS' COMMENTS:

Reviewer #1 (Remarks to the Author):

The authors have done a very good job of addressing my concerns.

Reviewer #2 (Remarks to the Author):

This is an original, interesting and relevant study, and Axelsson and colleagues have answered in an adequate way to most of my critiques. There remain, however, a couple of points to be corrected:

1. Suppl. Fig. 5g shows a 1.2 fold-change increase in Mafa in Sox5 overexpressing cells as compared to controls. Surprisingly, this is described in Results, page 16, line 329, as a "... (100- to 600-fold increase) compared with control cells, without reaching statistical significance...". Please clarify or correct.
2. In Results, page 23, line 485, please remove "...and VPA both..." from the title, since the VPA experiments were removed from this part of the text.
3. In Supplementary Experimental Procedures, page 29, lines 256-259, please indicate the number of mice exposed to phlorizin or control treatment.
4. In Supplementary Experimental Procedures, page 35, line 542, add a "the": "...For immunohistochemistry THE following primary antibodies were used:..."

Reviewer #3 (Remarks to the Author):

The authors have satisfactorily addressed earlier concerns. However, the use of the term 'dedifferentiation' here appears unfounded as the T2D signature characteristics are quite distinct from those associated with this phenomenon in experimental models (ref 20). Thus, there is no induction of human Ngn3, Nanog or other developmental progenitor markers within T2D islets. Consequently, a better definition of the T2D signature seen here is 'immaturity' or 'loss of beta cell identity'.

Response to reviewers

Reviewer #1 (Remarks to the Author):

The authors have done a very good job of addressing my concerns.

We thank this reviewer for his/her positive attitude towards the manuscript.

Reviewer #2 (Remarks to the Author):

This is an original, interesting and relevant study, and Axelsson and colleagues have answered in an adequate way to most of my critiques. There remain, however, a couple of points to be corrected:

We thank the reviewer for his/her constructive comments throughout the review process.

1. Suppl. Fig. 5g shows a 1.2 fold-change increase in Mafa in Sox5 overexpressing cells as compared to controls. Surprisingly, this is described in Results, page 16, line 329, as a “...(100- to 600-fold increase) compared with control cells, without reaching statistical significance...”. Please clarify or correct.

This 100- to 600-fold increase refers to the fold change of Sox5 expression upon overexpression and not to MafA levels. We realize that there is room for misinterpretation, and have now specified that the increase refers to Sox5 levels.

2. In Results, page 23, line 485, please remove “...and VPA both...” from the title, since the VPA experiments were removed from this part of the text.

This has now been corrected.

3. In Supplementary Experimental Procedures, page 29, lines 256-259, please indicate the number of mice exposed to phlorizin or control treatment.

This information has been added to the figure legend of Figure 5.

4. In Supplementary Experimental Procedures, page 35, line 542, add a “the”: “...For immunohistochemistry THE following primary antibodies were used:...”

This has now been added.

Reviewer #3 (Remarks to the Author):

We thank the reviewer for his/her helpful and insightful comments in the review process.

The authors have satisfactorily addressed earlier concerns. However, the use of the term 'dedifferentiation' here appears unfounded as the T2D signature characteristics are quite distinct from those associated with this phenomenon in experimental models (ref 20). Thus, there is no induction of human Ngn3, Nanog or other developmental progenitor markers within T2D islets. Consequently, a better definition of the T2D signature seen here is 'immaturity' or 'loss of beta cell identity'.

We have included a statement on this in the discussion on page 29.